# Broadband Three-Mode Tunable Metamaterials Based on Graphene and Vanadium Oxide

**DOI:** 10.3390/nano15201572

**Published:** 2025-10-16

**Authors:** Hao Wen, Shouwei Wang, Yiyang Cai, Zhuochen Zou, Zheng Qin, Tianyu Gao

**Affiliations:** 1School of Intelligent Manufacturing, Nanjing University of Science and Technology, Nanjing 210094, China; wenhao123@njust.edu.cn (H.W.); wangshouwei@njust.edu.cn (S.W.); 2Sino-French Engineer School, Nanjing University of Science and Technology, Nanjing 210094, China; caiyiyang@njust.edu.cn (Y.C.); dmicmo@njust.edu.cn (Z.Z.)

**Keywords:** terahertz, metamaterial, graphene, vanadium dioxide, broadband absorption, high transmissivity, dynamic modulation, impedance matching

## Abstract

Terahertz waves have great potential for applications in security imaging, wireless communication, and other fields, but efficient and tunable terahertz-absorbing devices are the key to their technological development. In this paper, a tunable terahertz metamaterial based on graphene and vanadium dioxide materials is proposed. When the vanadium dioxide conductivity is 1.6 × 10^5^ S/m and the Fermi energy level of graphene is 0.75 eV, the metamaterial exhibits high absorptivity exceeding 90% in ultra-broadband of 2.05–14.03 THz; when the Fermi energy level of graphene is adjusted to 0 eV, the high absorption wavelength range narrowed to 4.07–13.80 THz; when the vanadium dioxide conductivity is adjusted to 200 S/m, the metamaterial exhibits high transmissivity exceeding 80% in the wavelength range up to 15 THz. Additionally, the metamaterial is insensitive to polarization angles and incident angles, allowing it to adapt to changes in the angle of incidence and polarization in practical applications. The metamaterial has potential applications in optical switches, electromagnetic wave stealth devices, and filtering devices.

## 1. Introduction

As a type of electromagnetic radiation between microwave and infrared, the terahertz wave, with its unique advantages of strong penetrability, low energy, and rich spectral information, has shown great application potential in the fields of security imaging [1,2], wireless communication [3,4,5], solar energy utilization [6,7] and biomedical applications [8,9]. However, the further development of terahertz technology is constrained by the performance of key functional devices. Highly efficient and tunable terahertz metamaterial devices are just among them.

Metamaterials, a class of composite materials with artificially designed microstructures, can achieve an unattainable electromagnetic response of natural materials by modulating the geometrical parameters of the unit structure, providing an ideal platform for constructing high-performance terahertz metamaterial devices. In 2008, Landy et al. [10] first proposed the concept of metamaterial absorber. Later in 2008, Tao et al. [11] designed a terahertz metamaterial structure through parameter optimization and achieved 97% absorptivity at 1.6 THz. In 2009, Wen et al. [12] designed a terahertz metamaterial structure that achieved exceeding 90% absorptivity from 0.8 THz to 1.2 THz. Currently, the research on terahertz metamaterials primarily focuses on expanding the absorption bandwidth, enhancing the tunable range and increasing the absorptivity. Hu et al. [13] developed a tunable metamaterial structure capable of switching among three absorption modes: low-frequency single-band, high-frequency single-band, and dual-band absorption. And Geng et al. [14] designed a metamaterial structure based on graphene and vanadium dioxide(VO_2_), which can realize the switching function of broadband absorption in the low frequency range, high frequency range, and low frequency to high frequency range.

Graphene, a two-dimensional carbon material with a single atomic layer thickness, exhibits excellent electrical tunability due to its unique Dirac cone energy band structure. The Fermi energy level of graphene can be effectively modulated through electric field application or chemical doping, enabling dynamic control of its terahertz electromagnetic properties [15,16]. This unique characteristic enables graphene–metamaterial hybrid structures to effectively overcome the performance limitations of conventional metamaterial devices. Li et al. [17] developed a globally gate-tunable graphene metasurface capable of dynamic terahertz wavefront reconfiguration through Fermi level modulation. This platform enables beam steering with ±30° reflection angle switching in the 0.3–1.2 THz range, and polarization state conversion from linear to circular polarization. Zhu et al. [18] demonstrated a graphene-based metamaterial structure that achieves tunable absorption peaks between 1.01 THz and 1.86 THz through Fermi level modulation, with peak absorptivity exceeding 99%. Yang et al. [19] utilize graphene to design a metamaterial structure, achieving an absorptivity exceeding 99% for the two peaks at 3.85 THz and 5.04 THz. The peak shifts can be controlled by adjusting the Fermi energy level of graphene, allowing for precise tuning of the peak shifts. The tuning function of the peak shift can be achieved by adjusting the Fermi energy level of graphene. Graphene exhibits tunable properties but faces limitations in achieving high-frequency broadband performance. The addition of VO_2_ compensates for this deficiency, enhancing the absorption in the high-frequency band.

VO_2_ is an outstanding phase change material with outstanding phase change properties insulator-to-metal phase transition occurs near a critical temperature of approximately 68 °C, exhibiting a reversible insulator–metal phase change. The phase change is reversible when the temperature is below the critical temperature of the insulator–metal phase change. When the temperature is lower than the phase transition point, VO_2_ presents insulating state, the lattice structure is monoclinic phase, and shows high-transmissivity characteristics to terahertz waves; when the temperature is increased to above the phase transition point, the lattice structure is transformed into a metallic state of the rutile phase, the electrical conductivity is increased drastically, and the reflecting and absorbing characteristics of the terahertz waves are significantly changed, and the dramatic change in the electromagnetic properties makes it an ideal material for the dynamic control of terahertz band [20,21]. Ge et al. [22] designed a metamaterial structure that can switch between four narrow-band absorption peaks and high-transmissivity transmission. In the transmission mode, 98.2% transmissivity is achieved at 6.2 THz and the reflectivity does not exceed 3% over the operating frequency range. Wang et al. [23] regulated the conductivity of VO_2_ through temperature based on the phase transition characteristics of VO_2_ and realized the switch between absorptivity exceeding 90% and total reflection (absorptivity not exceeding 4%) in the range of 3.01–7.27 THz. Yang et al. [24] developed a VO_2_-based metamaterial that achieves broadband absorption (exceeding 90%) spanning 2.81–7.71 THz. The absorptivity can be dynamically tuned from exceeding 90% to not exceeding 5% across this frequency band through VO_2_ conductivity modulation. However, the existing research has achieved rich performance in adjusting the absorption rate. There are few reports on the switching between broadband absorption and transmission functions, despite its significant application potential.

Here we propose a terahertz metamaterial structure that can switch between broadband absorption and transmission using graphene and VO_2_ materials, due to the limited applications of metamaterial devices that currently achieve broadband absorptivity tuning. Different from the metal dielectric layer used in traditional metamaterial structures, we utilize the structure of VO_2_-SiO_2_-VO_2_-SiO_2_-Graphene (continuous layer–dielectrics –resonator layer– dielectrics resonator layer), which is more conducive to the metamaterial realizing the function of transmission. The simulation results show that by controlling the conductivity of VO_2_ and the Fermi energy level of graphene, the metamaterial can be switched between 2.05–14.03 THz long broadband absorption (Mode 1), 4.07–13.80 THz short broadband absorption (Mode 2), as well as transmission (Mode 3), the absorptivity exceeds 90%, and the average transmissivity is up to 88% in all transmission modes. In addition, the metamaterial is insensitive to the polarization angle of the incident light. The physical mechanism of the metamaterial can be explained by the impedance matching principle, and its working principle is further analyzed in this paper through an electric field distribution diagram. From the simulation results, it is evident that the metamaterial has potential applications in optical switching, stealth, and filtering, among others.

## 2. Design and Method

In the metamaterial structure proposed in this paper, the Fermi energy level of graphene and the conductivity of VO_2_ can be adjusted to achieve a tuning function that dynamically adjusts the broadband absorption range. The specific unit structure is shown in Figure 1a. The metamaterial structure consists of five layers, from bottom to top: a continuous VO_2_ layer, a silicon dioxide dielectric layer, a VO_2_ resonator layer, a silicon dioxide dielectric layer, and a graphene resonator layer. The thickness of each layer is *t*_1_, *h*_1_, *t*_2_, *h*_2_, respectively. The VO_2_ resonator layer is formed by cutting three concentric rings: all three rings have the same thickness *c*; the ring closest to the center is uncut and has an inner diameter *r*_1_; the middle ring has openings in the four directions 45°, 135°, −135°, −45°, the width of the openings is *b*, and the inner diameter of the ring is *r*_2_; the outer ring has openings in the four directions 0°, 90°, 180°, −90°, the width of the openings is *a*, and the inner diameter is *r*_3_. The four corners of the top graphene patterned layer consist of four quarter circles with radius *r*_4_; the pattern in the center consists of a cross adding four half-circles with radius *r*_5_ in the four directions, where the lengths of the crosses in the transverse and longitudinal directions are *d*. The cell period is *p*. The pattern in the center consists of a cross adding four half-circles with radius *r*_5_ in the four directions.

The concentric ring VO_2_ structure enables multi-resonance coupling and symmetry optimization, achieving ultra-wideband, polarization-insensitive terahertz absorption. Its outer and middle ring openings form a parallel slit capacitor; when terahertz waves hit vertically with the electric field parallel to the slit, the structure acts as an LC circuit, triggering strong resonance to enhance absorption via slit electric field, metal ring eddy current loss, and Joule heat. The graphene cross-quarter-circle combination realizes “peak complementarity”, enhances absorption through local electric field superposition, enables 2–4 THz broadband absorption, packs more resonators in limited area, and interacts with terahertz waves in multiple ways to expand bandwidth.

Graphene has the advantages of continuous fine regulation in terahertz band, very fast response, and low static loss, but needs to continuously apply the bias voltage to maintain the regulatory state; VO_2_ can be achieved through temperature regulation of the phase transition, and after the phase transition without external stimulation to maintain the state, there is low insulating loss, but the metal phase loss is higher, indicating the lack of continuity of regulation. After the combination of the two, not only can VO_2_ achieve “coarse tuning switch”, that is, from the broadband absorption of Mode 1 to switch to the high-transmissivity transmission of Mode 3, but also with the help of graphene to complete the “fine tuning compensation”, that is, the absorption bandwidth between mode 1 and Mode 2 length of the adjustment.

We define the frequency band range with absorption rate greater than 90% as the absorption bandwidth, and we select the absorption bandwidth of VO_2_ in the metallic state as the main optimization goal and the average transmittance of VO_2_ in the dielectric state as the secondary optimization target. After several rounds of simulation optimization, we obtain the optimal structural parameters: *r*_1_ = 1.5 μm, *r*_2_ = 4.5 μm, *r*_3_ = 7.5 μm, *r*_4_ = 5 μm, *r*_5_ = 3 μm, *a* = 2 μm, *b* = 1 μm, *c* = 1.5 μm, *d* = 10 μm, *t*_1_ = 0.3 μm, *t*_2_ = 0.25 μm, *h*_1_ = 4.25 μm, *h*_2_ = 7.7 μm, *p* = 19 μm.

Graphene is often considered as an infinitely thin surface that can be defined using the surface conductivity *σ*. According to Kubo’s equation [25,26], the conductivity of graphene can be described as follows:
(1)$$\sigma_{\mathrm{gra}}=\sigma_{\mathrm{intra}}+\sigma_{\mathrm{inter}}$$
(2)$$\sigma_{\mathrm{intra}}=\frac{2e^{2}k_{B}T}{\pi \hbar^{2}}\frac{i}{\omega +\frac{i}{\tau }}\ln\left[2\cosh\left(\frac{E_{f}}{2k_{B}T}\right)\right]$$
(3)$$\sigma_{\mathrm{inter}}=\frac{e^{2}}{4\hbar^{2}}\left[\frac{1}{2}+\frac{1}{\pi }\arctan\left(\frac{\hbar \omega -2E_{f}}{2}\right)-\frac{i}{2\pi }\ln\frac{(\hbar \omega +2E_{f}{)}^{2}}{(\hbar \omega +2E_{f}{)}^{2}+4(k_{B}T{)}^{2}}\right]$$

In the formulae,
$E_{f}$,
T,
e,
τ,
ω,
ℏ, and
$k_{B}$ denote the Fermi energy level of graphene, ambient temperature, electron charge, carrier relaxation time, angular frequency of incident light, reduced Planck’s constant, and Boltzmann’s constant, respectively. Since the frequency range investigated in this paper is within the terahertz band, the Fermi energy levels of graphene satisfy the relation:
$E_{f}\gg \frac{\hbar \omega }{2}$; and in this frequency range, the effect of interband jumps on the conductivity of graphene is negligible due to the bubbleley blocking effect [27]. In addition to this,
$E_{f}\gg k_{B}T$. Based on all the conditions mentioned above, a simplified Drude Model can be used to describe the conductivity of graphene in the study of this paper [28]:
(4)$$\sigma_{\mathrm{gra}}=\frac{e^{2}E_{f}}{\pi \hbar^{2}}\frac{i}{\left(\omega +\frac{i}{\tau }\right)}$$

From Equation (4), it can be seen that *σ_gra_* depends on *E_f_*, *τ*, and *ω* since *e* and the *ℏ* are not affected by the experimental environment. Therefore, we can alter the Fermi energy level of graphene through chemical doping or applying a bias voltage to adjust the chemical potential, thereby modifying *σ_gra_*. Among them, the adjustment range of graphene Fermi energy level is generally from 0 eV to 0.9 eV. In this paper, *τ* = 0.1 ps is selected.

The permittivity of VO_2_ in the terahertz band can be described by the Drude Model [29]:
(5)$$\varepsilon (\omega )=\varepsilon_{\infty }-\frac{\omega_{p}^{2}(\sigma )}{\omega (\omega \;+i\gamma )}$$

In the equation, *ε*_∞_ = 12 corresponds to the permittivity at infinity frequency, and *γ* = 5.75 × 10^13^ rad/s denotes the damping frequency. The relation between the plasma frequency *ω_p_* and the conductivity *σ* can be expressed as [30,31]
(6)$$\omega_{p}^{2}(\sigma )=\frac{\sigma }{\sigma_{0}}\omega_{p}^{2}(\sigma_{0})$$ where *σ*_0_ = 3 × 10^5^ S/m, *ω_p_*(*σ*_0_) = 1.4 × 10^15^ rad/s. By adjusting the temperature, the conductivity *σ* of VO_2_ can be varied in the range of 200 S/m to 2 × 10^5^ S/m. In the experiment, VO_2_ is in the insulating state at *σ* = 200 S/m and in the metallic state at *σ* = 2 × 10^5^ S/m.

We performed simulations using CST Microwave Studio, applying periodic boundary conditions along the X and Y axes while setting open boundary conditions for the Z-axis. The electromagnetic wave was incident normally on the metamaterial surface along the negative Z-direction. Using a single-unit cell approach, we obtained the S-parameters to accelerate computations while maintaining accuracy. From these parameters, we calculated the reflectivity (R) and transmissivity (T). Eventually, the absorptivity can be described by the following equation [32,33]:
(7)$$A(\omega )=1-R(\omega )-T(\omega )=1-|S_{11}(\omega ){|}^{2}-|S_{21}(\omega ){|}^{2}$$

In the equation,
$R(\omega )=|S_{11}(\omega ){|}^{2}$ denotes the reflectance of the device,
$T(\omega )=|S_{21}(\omega ){|}^{2}$ denotes the transmissivity of the device,
$S_{11}(\omega )$ denotes the reflection coefficient and
$S_{21}(\omega )$ denotes the transmission coefficient. It is worth noting that at high temperatures, the underlying VO_2_ can be considered as a metallic layer due to the metallic conductivity of the VO_2_. In this case, since the thickness of this layer is significantly greater than the skin depth of the terahertz wave, we can assume that the transmittance of the device is zero at high temperatures.

## 3. Results and Discussion

We first set the Fermi energy level of graphene to 0 eV and cut the concentric rings in steps, and the spectral absorptivity before and after cutting is shown in Figure 2. From the figure, it can be seen that in the frequency range of 4.07–13.80 THz, the absorptivity of a single concentric ring (dark blue line) can exceed 90% in only a few frequency ranges, which has not reached the expected effect; after cutting the outer ring (gray line), the absorptivity has come close to exceeding 90% in the same frequency ranges, but there still exists a section of the absorptivity near 8 THz that do not exceed 90%; after continuing to cut the middle ring (red line), it can be seen that the absorptivity has exceeded 90% in the frequency range, achieving broadband absorption.

The absorption as well as transmission spectra of the metamaterial designed in this paper in three modes are shown in Figure 3. TM wave represents the linearly polarized light with electric field along the X-axis when the incident direction is the Z-axis; TE wave represents the linearly polarized light with electric field along the Y-axis when the incident direction is the Z-axis. As can be seen from Figure 3b–d, the curves obtained from the simulation are highly consistent when the incident wave is a TE wave and a TM wave in the three modes, respectively, indicating that the metamaterial is insensitive to the polarization angle of the incident light.

Figure 3a shows the absorption, reflection, and transmission spectra of the metamaterial in Mode 1 and Mode 2. When the VO_2_ conductivity is set to 1.6 × 10^5^ S/m and the Fermi energy level of graphene is set to 0.75 eV, the metamaterial wave will operate in Mode 1. Simulation results demonstrate that Mode 1 achieves broadband absorption across 2.05–14.03 THz (bandwidth: 11.98 THz), with exceeding 90% absorptivity and a 94% average absorptivity. In this mode, we obtained high absorptivity at frequencies of 2.31 THz, 3.35 THz, 9.63 THz, 11.24 THz, and 13.54 THz, with corresponding absorptivity values of 94.6%, 96.1%, 96.6%, 95.8%, and 99.7%. When the VO_2_ conductivity is set to 1.6 × 10^5^ S/m and the Fermi energy level of graphene is set to 0 eV, the metamaterial will operate in Mode 2. Mode 2 achieves broadband absorption from 4.07 to 13.80 THz, with a bandwidth of 9.73 THz. The absorptivity exceeds 90% and the average absorptivity is 95%. Mode 2 exhibits high absorptivity peaks at 5.45 THz, 9.22 THz, 11.08 THz, and 13.25 THz frequencies, corresponding to absorptivity of 97.0%, 97.9%, 96.7%, and 99.9%. Figure 3d demonstrates the spectral transmissivity diagram of the metamaterial in Mode 3. When the VO_2_ conductivity is set to 200 S/m and the Fermi energy level of graphene is set to 0 eV, the metamaterial will operate in Mode 3. At this time, the metamaterial is dominated by the transmission function, with transmissivity exceeding 80%, an average transmissivity of 88% and absorptivity not exceeding 2%. In this mode, transmission is achieved along with almost no absorption of the incident wave.

Figure 4 demonstrates the effect of VO_2_ conductivity on the absorptivity as well as the transmissivity of the metamaterial at the graphene Fermi energy level of 0 eV. As can be seen from Figure 4a,b, as the VO_2_ conductivity decreases from 1.6 × 10^5^ S/m to 200 S/m, the metamaterial gradually shifts from broadband efficient absorption in Mode 2 to transmission in Mode 3, and the transition from exceeding 90% absorptivity to exceeding 80% transmissivity is realized, which opens up the possibility of the application of broadband absorbing materials in optical switches.

Figure 5 shows the effect of adjusting the Fermi level of graphene from 0.5 eV to 0.9 eV on the absorptivity of the metamaterial when the conductivity of VO_2_ is set to 1.6 × 10^5^ S/m. With the increase in the Fermi level of graphene, the absorptivity of the metamaterial increases in the 2–3 THz band but decreases in the 8–9 THz band. When the Fermi energy level of graphene is 0.75 eV (pale yellow line), the comprehensive absorptivity performance of the metamaterial reaches its best. This not only ensures that the absorptivity of the metamaterial remains exceeding 90% during the tuning process but also enables the metamaterial to achieve dynamic tuning of the absorption bandwidth.

As can be seen from the comparison in Table 1, the advantage of the metamaterial proposed in this paper over previous broadband absorption studies is that it can obtain two switchable ultra-broad absorption bands of 11.98 THz and 9.73 THz while guaranteeing an absorptivity exceeding 90% [34,35,36,37,38,39]. Although the switching function between absorption and transmission has been investigated using VO_2_ materials (e.g., reference [22] achieved switching between multiband absorption and high-transmissivity transmission; references [38,39] achieved switching between broadband absorption and transmission), there is a lack of research on switching between ultra-broadband absorption and high-transmissivity modes. The proposed metamaterial structure can realize the mode switching between two ultra-wideband absorption and the function switching between ultra-wideband absorption and broadband transmission, with an average transmissivity of 88% in the operating frequency range, which effectively fills this research gap. This rapid switching (VO_2_ phase transition response time up to nanoseconds, graphene electrical regulation for picoseconds) can be used for terahertz optical switching arrays, support for optical computing, optical interconnections, and other areas of high-speed signal processing.

We can utilize the impedance matching theory to explain the physical mechanism behind the perfect absorption of this metamaterial. The formula for impedance matching is shown below [40]:
(8)$$Z=\sqrt{\frac{(1+S_{11}{)}^{2}-S_{21}^{2}}{(1-S_{11}{)}^{2}-S_{21}^{2}}}$$ where
$S_{11}(\omega )$ denotes the reflection coefficient and
$S_{21}(\omega )$ denotes the transmission coefficient.

We have investigated impedance matching and given the real and imaginary parts of the impedance matching of the metamaterial in the Mode 1 and Mode 2 operating bands, as shown in Figure 6. Impedance matching requires that the relative impedance of the material tends to be close to 1 in the real part and 0 in the imaginary part; this is conducive to the reduction in reflection so that the energy can be better transmitted or absorbed. The imaginary part reflects the energy storage properties of the material; the smaller the value of the imaginary part, the weaker the resistance of the material will be, and the characteristics of the material will be closer to a pure resistance, which is consistent with the requirement of impedance matching for the imaginary part [41,42].

From Figure 6a, it can be seen that the real part (black line) does not converge to 1 (upper dashed line) and the imaginary part (red line) does not converge to 0 (lower dashed line) in the range of 0.1–2 THz. This indicates that the relative input impedance does not match with the free-space impedance, which makes the terahertz wave incident to the metamaterial device reflected at the interface, and the metamaterial device’s absorption performance is poor, which is consistent with the simulation results obtained in Figure 3b. After 2 THz, the real part shows a rising trend and tends to 1, and the imaginary part shows a decreasing trend and tends to 0, and the effective impedance of the metamaterial matches with the free-space impedance, which achieves an effective absorption in the terahertz band. As can be seen in Figure 6b, near 3 THz, the real part is much larger than the imaginary part, indicating that the real part of the input impedance deviates from the corresponding real part of the free-space impedance, which results in strong reflection of the incident terahertz wave on the surface of the metamaterial device, and the absorption performance of the metamaterial device decreases, which corresponds to that of the absorption near 3 THz shown in Figure 3c. After 3 THz, the real part shows a decreasing trend and tends to 1, and the imaginary part shows an increasing trend and tends to 0. The effective impedance matches the free-space impedance, and the wave-absorbing performance of the metamaterial device rises.

To further illustrate the physical mechanism by which this metamaterial achieves broadband efficient absorption, we also plotted the electric field distributions of several peaks of the metamaterial when it is operated under Model 1, as shown in Figure 7. From Figure 7a,b, at the low-frequency absorptivity peaks of 2.31 THz and 3.35 THz, it can be seen from the figure that the electric field is mainly concentrated in the edge part of the metamaterial structure, which shows a more obvious edge enhancement effect. This is because the boundary conditions at the edges make the electric field strongly coupled and localized when the low-frequency electromagnetic wave interacts with the metamaterial structure [43]. For the metamaterial, this strong electric field distribution at the edges leads to enhanced absorption of low-frequency electromagnetic waves, and the electromagnetic energy is converted into other forms of energy through the resonance of the edge structure with the electromagnetic waves. At the mid-frequency absorptivity peak at 9.63 THz, as shown in Figure 7c, the electric field distribution begins to change. The electric field has a wider distribution inside the structure and is no longer concentrated only at the edges. This indicates that as the frequency increases, the internal microstructure begins to have a significant effect on the electric field distribution, with the overall structure of the metamaterial being involved in the interaction with the electromagnetic wave. From Figure 7d,e, at the high-frequency absorptivity peaks of 11.24 THz and 13.54 THz, the electric field distribution is more complex and exhibits characteristics of localization in multiple regions. The electric field has strong distribution regions in different parts of the metamaterial structure, indicating that the high-frequency electromagnetic wave resonantly couples with multiple microstructural features of the metamaterial. High-frequency electromagnetic waves can excite more resonant modes of different scales and forms in metamaterials, which makes the electric field form strong localization in multiple regions, which is essential for realizing the absorption of electromagnetic waves in a wide bandwidth, and expanding the working frequency band of the metamaterial through the absorption of electromagnetic waves of different frequencies in different regions. Combined with the results given in Figure 6a, it can be seen that near the above frequencies, the real part is very close to 1 and the imaginary part is very close to 0. The input impedance matches with the spatial impedance, and the metamaterials can achieve a high absorption rate. This is consistent with the results demonstrated in Figure 7 that the electric field is locally strong at each frequency, with significant energy concentration and high absorption efficiency of the metamaterial.

Since terahertz waves can be incident from different angles in practical applications, it is essential for metamaterial devices to have good wide-angle and polarization insensitivity. Figure 8 shows the absorptivity of TE and TM waves by three modes of the metamaterial at different incidence angles. From Figure 8a,b, it can be seen that under the TE polarization condition, the metamaterial can achieve exceeding 90% absorptivity in Mode 1 for the incident waves with about 2–14 THz incidence angles from 0° to 50°. Under TM polarization, the absorptivity of metamaterials to incident waves with incident angles of 0° to 70° in the same frequency range exceeds 90%. From Figure 8c,d, it can be seen that the metamaterials can exceed 90% absorptivity of incident waves in the range of 4–13.5 THz for incident angles of 0° to 60° under both TE and TM polarization conditions in Mode 2. Figure 8e demonstrates that under TE polarization conditions, the metamaterial can exceed 80% transmissivity for incident waves with an incident angle of 0–30°at about 0.1–1.5 THz, 6.5–8 THz, and 13–15 THz in Mode 3. Figure 8f demonstrates that the metamaterial can achieve exceeding 90% transmissivity for incident waves with incident angles of approximately 0–70° at 0.1–15 THz under TM polarization conditions in Mode 3. It can be seen that the metamaterial exhibits good polarization insensitivity as well as wide-angle absorption, which enables it to adapt to changes in incident angle and polarization angle in practical applications.

We investigated the influence of geometry on the performance of metamaterial devices, including the optimizing of the period *p*, the thickness of the VO_2_ pattern layer *t*_2_, and the relevant parameters of the graphene pattern layer *d*, *r*_4_, and *r*_5_, as shown in Figure 9 and Figure 10. Figure 9a,b demonstrate the effect of different *p* on the absorptivity of Mode 1 and Mode 2, respectively. From the figure, it can be seen that the absorptivity of the metamaterial in the mid-frequency region of 6–9 THz decreases gradually with the gradual increase of *p* from 19 μm in Mode 1, and the absorptivity in some frequency ranges does not exceed 90%, which affects the overall broadband absorption effect. The absorption bandwidth of the metamaterial in Mode 1 reaches the maximum value of 11.98 THz when *p* = 19 μm (dark red line). The overall absorptivity of the metamaterial in the operating frequency range in Mode 2 exceeds 90% only when *p* = 19 μm and *p* = 20 μm (orange line). The absorption bandwidth is 9.73 THz with an average absorptivity of 95.07% when *p* = 19 μm; the absorption bandwidth is 9.09 THz with an average absorptivity of 94.08% when *p* = 20 μm. In summary, *p* = 19 μm is chosen as the final parameter. Figure 9c,d show the effect of different *t*_2_ on the absorptivity for Mode 1 and Mode 2, respectively. It can be seen that although at *t*_2_ = 0.3 μm (pale yellow line), the absorption of the metamaterial in most of the frequency range in Mode 1 is better than the other results, the absorptivity near 12 THz in Mode 2 does not exceed 90%, which affects the overall broadband absorption effect. Whereas at *t*_2_ = 0.25 μm (orange line), the absorptivity of the metamaterial in the overall frequency range of both Mode 1 and Mode 2 exceeds 90%. Moreover, the average absorptivity of the metamaterial in the operating frequency range reaches 95.07% when *t*_2_ = 0.25 μm, which is better than the average absorptivity of 94.2% when *t*_2_ = 0.3 μm. Based on the above, *t*_2_ = 0.25 μm is chosen.

Figure 10a shows the effect of different *r*_4_ on the absorptivity of Mode 1. From the figure, it can be seen that the absorption bandwidth of the metamaterial is the longest when *r*_4_ =7 μm (purple line), but its absorptivity decreases significantly near 3 THz. Only when *r*_4_ = 5 μm, the overall absorptivity of the metamaterial in the operating frequency range exceeds 90% and the absorption bandwidth of 11.98 THz reaches the maximum. Therefore, *r*_4_ = 5 μm is chosen. Figure 10b demonstrates the effect of different *d* on Mode 1 absorptivity. Only when *d* = 10 μm (pale yellow line) and *d* = 11 μm (green line), the overall absorptivity of the metamaterial exceeds 90% in the operating frequency range. The absorption bandwidth is 11.94 THz when *d* = 11 μm and 11.98 THz when *d* = 10 μm. So, *d* = 10 μm is chosen as the final parameter. Figure 10c demonstrates the effect of different *r_5_* on the absorptivity of Mode 1. Only when *r*_5_ = 2.5 μm (orange line) and *r*_5_ = 3 μm (pale yellow line), the overall absorptivity of the metamaterial exceeds 90% in the operating frequency range. The absorption bandwidth is 11.77 THz when *r*_5_ = 2.5 μm and 11.98 THz when *r*_5_ = 3 μm. So *r*_5_ = 3 μm is selected.

## 4. Conclusions

In this paper, we propose a terahertz metamaterial design based on graphene as well as VO_2_ materials. The structure consists of five layers: the bottom VO_2_ layer, the silicon dioxide dielectric layer, the VO_2_-patterned layer, the silicon dioxide dielectric layer, and the top graphene-patterned layer. Through simulation, we conclude that it is can operate in three different modes, which can achieve long broadband absorption in the frequency range of 2.05–14.03 THz with an absorptivity exceeding 90%, shorter broadband absorption in the frequency range of 4.07–13.80 THz with an absorptivity exceeding 90%, and high transmission in the frequency range of 0.1–15 THz with an average transmittance of 88%. Its core advantage lies in its ability to switch between ultra-broadband absorption and high-transmissivity transmission. To elucidate the physical mechanism of metamaterial operation, we have presented and analyzed the real and imaginary parts of the impedance matching of the metamaterial in the operating frequency bands of Mode 1 and Mode 2, based on the principle of impedance matching. Additionally, we plot the electric field distribution of several peaks of the metamaterial when it operates under Model 1, further illustrating the physical mechanism by which the metamaterial operates. To verify that the metamaterial exhibits good wide-angle absorption and polarization insensitivity, we present and analyze the absorption of the metamaterial in various modes at different incident angles and polarization angles. Finally, the influence of structural parameters on the performance of the metamaterial is equally important. We give the simulation optimization process for each structural parameter separately, and the optimal structural parameter is finally obtained.

## Figures and Tables

**Figure 1 nanomaterials-15-01572-f001:**
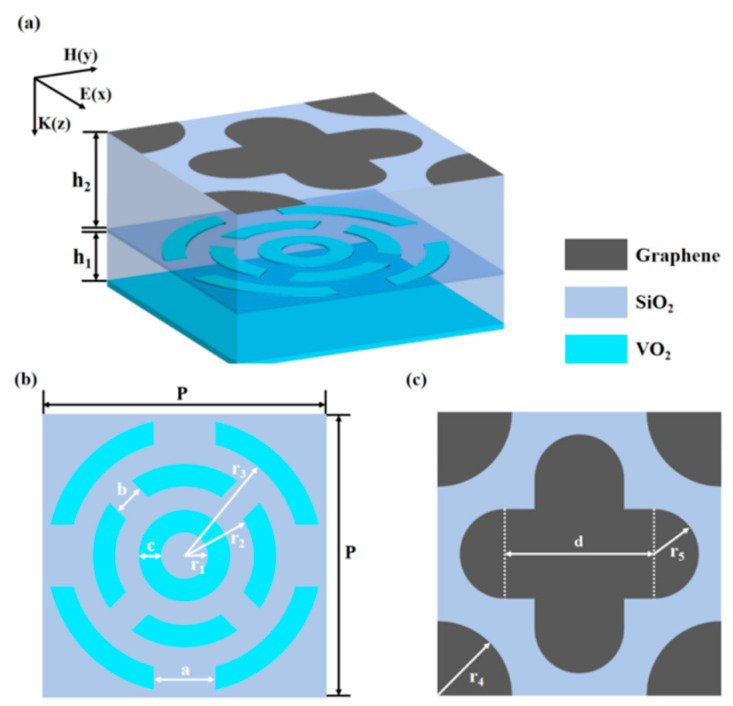
(**a**) Metamaterial 3D structure; (**b**) VO_2_-patterned layer structure; (**c**) Graphene-patterned layer structure.

**Figure 2 nanomaterials-15-01572-f002:**
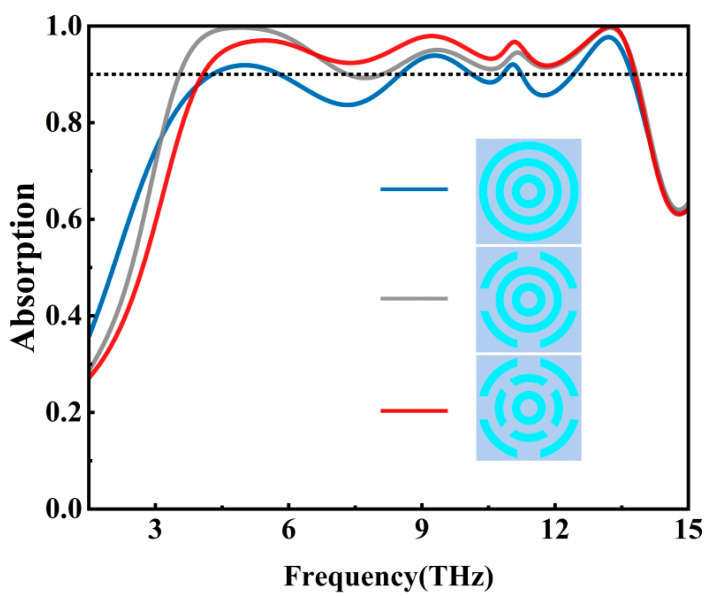
Spectral absorptivity of different VO_2_ patterned layers.

**Figure 3 nanomaterials-15-01572-f003:**
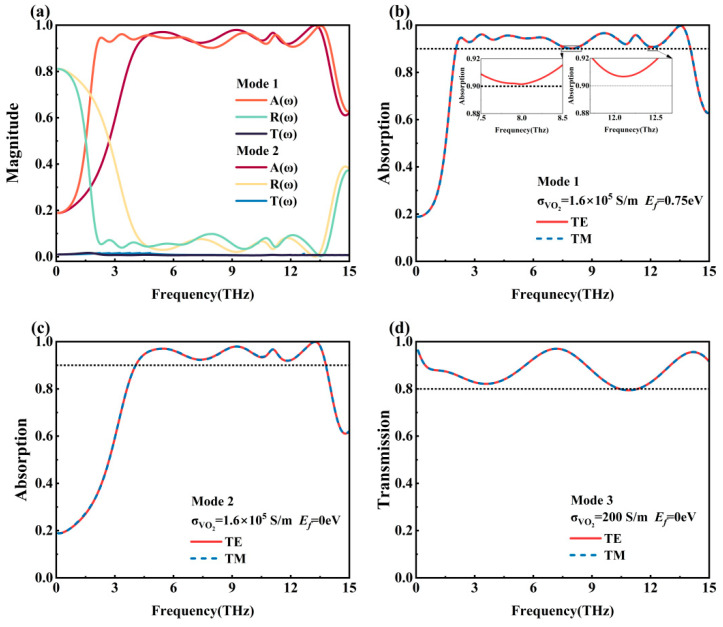
(**a**) Absorptivity, reflectivity and transmissivity spectra of Mode 1 and Mode 2; (**b**) absorptivity spectra of Mode 1 with different polarized incident waves; (**c**) absorptivity spectra of Mode 2 with different polarized incident waves; (**d**) transmissivity spectra of Mode 3 with different polarized incident waves.

**Figure 4 nanomaterials-15-01572-f004:**
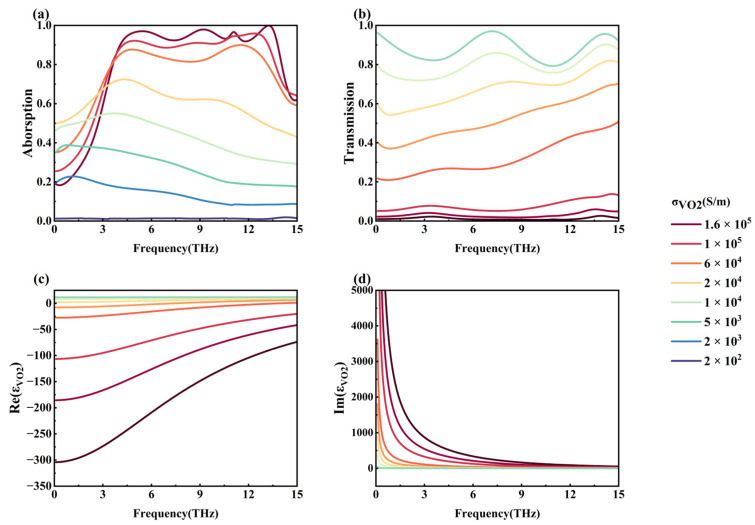
(**a**) Effect of VO_2_ conductivity on absorptivity; (**b**) effect of VO_2_ conductivity on transmissivity; (**c**) real part of VO_2_ dielectric constant; (**d**) imaginary part of VO_2_ dielectric constant.

**Figure 5 nanomaterials-15-01572-f005:**
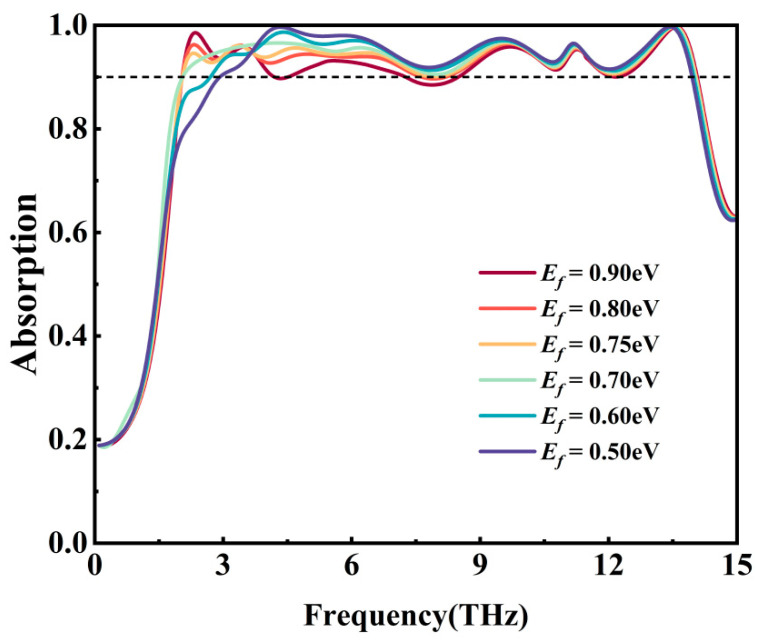
Influence of Fermi level of graphene on absorptivity.

**Figure 6 nanomaterials-15-01572-f006:**
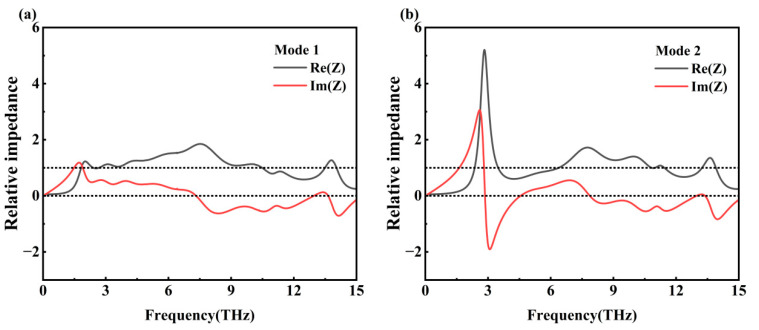
(**a**) Real and imaginary parts of impedance matching in Mode 1; (**b**) real and imaginary parts of impedance matching in Mode 2.

**Figure 7 nanomaterials-15-01572-f007:**
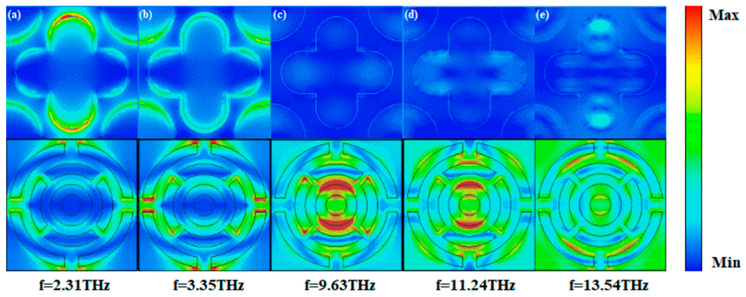
(**a**) Electric field distribution at 2.31 THz in Mode 1; (**b**) electric field distribution at 3.35 THz in Mode 1; (**c**) electric field distribution at 9.63 THz in Mode 1; (**d**) electric field distribution at 11.24 THz in Mode 1; (**e**) electric field distribution at 13.54 THz in Mode 1.

**Figure 8 nanomaterials-15-01572-f008:**
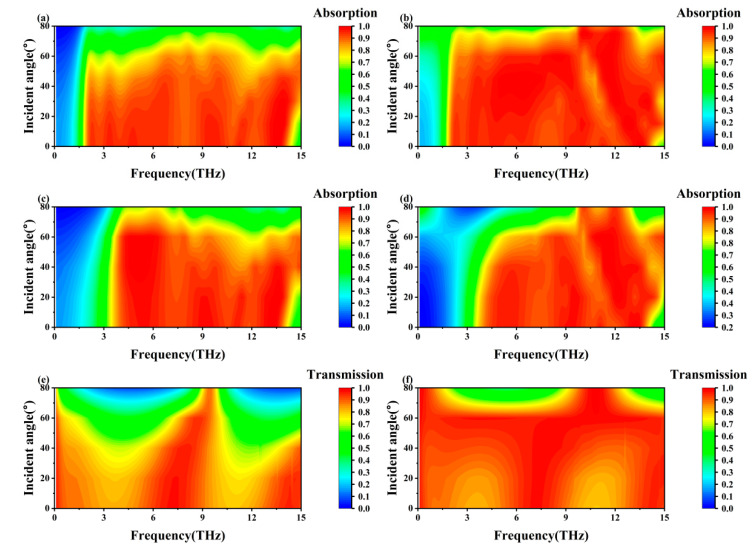
(**a**) Absorptivity of TE wave at different incidence angles for Mode 1; (**b**) absorptivity of TM wave at different incidence angles for Mode 1; (**c**) absorptivity of TE wave at different incidence angles for Mode 2; (**d**) absorptivity of TM wave at different incidence angles for Mode 2; (**e**) transmissivity of TE wave at different incidence angles for Mode 3; (**f**) transmissivity of TM wave at different incidence angles for Mode 3.

**Figure 9 nanomaterials-15-01572-f009:**
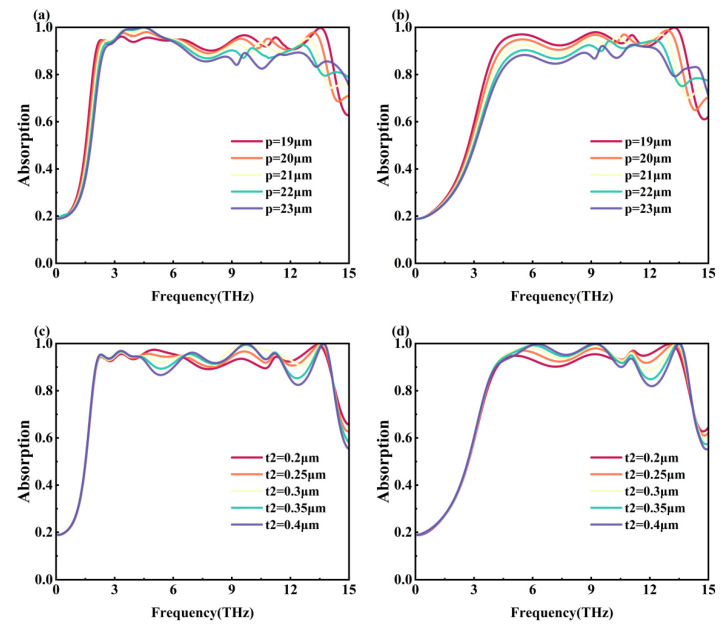
(**a**) Spectral absorptivity plots for different *p* in Mode 1; (**b**) spectral absorptivity plots for different *p* in Mode 2; (**c**) spectral absorptivity plots for different *t*_2_ in Mode 1; (**d**) spectral absorptivity plots for different *t*_2_ in Mode 2.

**Figure 10 nanomaterials-15-01572-f010:**
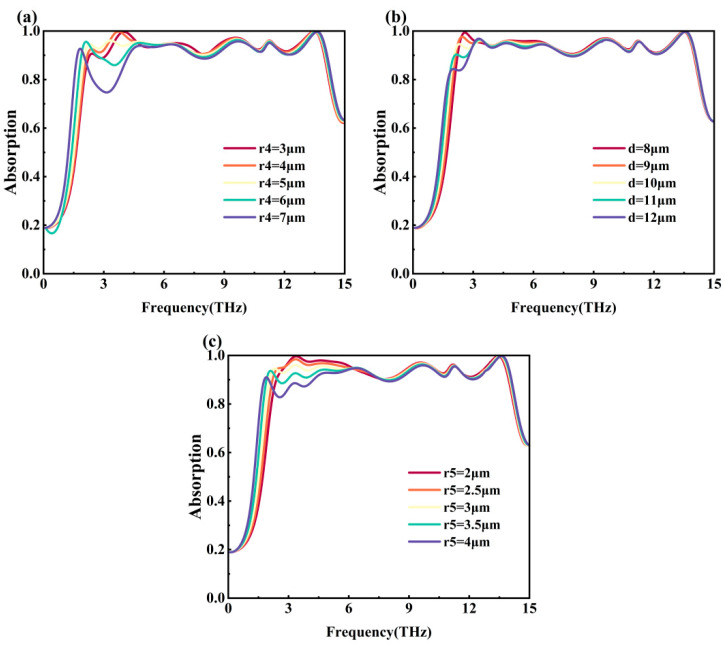
(**a**) Spectral absorptivity plots for different *r*_4_; (**b**) spectral absorptivity plots for different *d*; (**c**) spectral absorptivity plots for different *r*_5_.

**Table 1 nanomaterials-15-01572-t001:** Comparison of our designed metamaterial with previous similar studies.

References	Function (Absorption Band)	Broad Bandwidth > 90% (THz)	Performance	Average Transmissivity
[34]	Broadband and Multiband (three peaks)	1.56 (3.21–4.77)	Only absorption	_
[35]	Broadband	5.02 (1.11–6.13)	Only absorption	_
[36]	Broadband	3.22 (3.74–6.96)	Only absorption	_
[37]	Broadband and Multiband (two peaks)	3.37 (0.94–4.31)	Only absorption	_
[38]	Broadband	5.80 (3.60–9.40)	Absorption and transmission	86% (5.3–10 THz)
[39]	Broadband	4.39 (2.46–6.85)	Absorption and transmission	69.61% (0.01–2.32 THz) 69.79% (6.82–10.00 THz)
This work	Dual ultra-wideband	11.98 (2.05–14.03) 9.73 (4.07–13.80)	Absorption and transmission	88% (0.1–15 THz)

## Data Availability

The original contributions presented in this study are included in the article. Further inquiries can be directed to the corresponding authors.

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
