# Peer review of "Broadband Three-Mode Tunable Metamaterials Based on Graphene and Vanadium Oxide"

_nanomaterials, 2025, doi:10.3390/nano15201572_

Round 1

Reviewer 1 Report

Comments and Suggestions for Authors

The paper present the desing and simulation of metamaterial abrsorber in terahertz range using graphen and vabadium oxide. 

My major comment on the paper is the missing explanations on the desing principle, physical mechanism, simulation details and evaluation of the results. 

What is the design principle presented in figure 1? Please explain their physical mechanism or show appropriate reference explaining the physical mechanism. How is the bandwidth determined? how is the hight, h1 and h2 determined? Why do you make the slot of VO2 circle? How the graphen pattern designed? 

Simulation details are missing. What is the simulator used? Please show the direction of incident wave and definition of polarization for TM and TE simulation. Comparison of TM and TE in figure 4b,c,d may be just because the polarization synmetry. 

Simulation results in figure 8 should be discussed together with the physical mechanism. Advantage of the current design should be discussed in comparison to other references.

The relation between figure 7 and figure 4(a) should be explained, especially the feature at 3 THz.

For optimization of parameter, quantitative measure can be used for discussion, such as average transmission, minimum transmission and bandwidth.

Although experimantal evaluation is not addressed in the paper, you should discuss reliability of the simulation results. 

Followings are additional comments:

Line 80, the reference to [22] looks wrong.

The long explanation of figure 9 can be shortened.

Comments on the Quality of English Language

The long sentence in line 54-61 seems to be merged two sentences, and meaningless.

Sometimes strange wording can be seen such as, line 155 "calter" and line 255 "the closer" .

"approximate Planck's constant" should be "reduced Planck's constant"

Author Response

Manuscript ID: nanomaterials-3875084

Title: Broadband Three-Mode Tunable Metamaterials Based on Graphene and Vanadium Oxide

Corresponding Author: Zheng Qin, Tianyu Gao

Dear editors,

We would like to thank the reviewers for their thorough and careful reading of our manuscript, and  suggestions to improve our manuscript. We have carefully modified our manuscript strictly according to their suggestions, and hope that this revised manuscript can be accepted for publication.

We address in detail each point raised by the referee individually below; verbatim reviewer comment is

shown in blue, with our response in black under each comment. The corresponding changes in the

revised manuscript are marked in red for the deleted and highlight in yellow for the added content.

Sincerely yours,

Zheng Qin

***********************************************************************************

Response to the comments of the reviewers

***********************************************************************************

Reviewer 1:

The paper present the desing and simulation of metamaterial abrsorber in terahertz range using graphen and vabadium oxide. My major comment on the paper is the missing explanations on the desing principle, physical mechanism, simulation details and evaluation of the results. What is the design principle presented in figure 1? Please explain their physical mechanism or show appropriate reference explaining the physical mechanism. How is the bandwidth determined?

Response

We have supplemented the design principle, physical mechanism and simulation details accordingly. Especially in the physical mechanism, we combine the electric field distribution, impedance matching and simulation results to fully verify the physical mechanism of metamaterial operation and also the simulation results. Regarding the simulation details, we also give the relevant settings for simulation in the design methodology.

Update to the manuscript:

Result and discussion

We have investigated impedance matching and given the real and imaginary parts of the impedance matching of the metamaterial in the Mode 1 and Mode 2 operating bands, as shown in Fig. 6. Impedance matching requires that the relative impedance of the material tends to be close to 1 in the real part and 0 in the imaginary part; this is conducive to the reduction of reflection so that the energy can be better transmitted or absorbed. The imaginary part reflects the energy storage properties of the material; the smaller the value of the imaginary part, the weaker the resistance of the material will be, and the characteristics of the material will be closer to a pure resistance, which is consistent with the requirement of impedance matching for the imaginary part[39,40].

From Fig. 6(a), it can be seen that the real part (black line) do not converge to 1 (upper dashed line) and the imaginary part (red line) do not converge to 0 (lower dashed line) in the range of 0.1-2 THz. This indicates that the relative input impedance does not match with the free-space impedance, which makes the terahertz wave incident to the metamaterial device reflected at the interface, and the metamaterial device's absorption performance is poor, which is consistent with the simulation results obtained in Fig. 3(b).After 2 THz, the real part shows a rising trend and tends to 1, and the imaginary part shows a decreasing trend and tends to 0, and the effective impedance of the metamaterial matches with the free-space impedance, which achieves an effective absorption in the terahertz band. As can be seen in Fig. 6(b), near 3 THz, the real part is much larger than the imaginary part, indicating that the real part of the input impedance deviates from the corresponding real part of the free-space impedance, which results in strong reflection of the incident terahertz wave on the surface of the metamaterial device, and the absorption performance of the metamaterial device decreases, which corresponds to that of the absorption near 3 THz shown in Fig. 3(c). After 3 THz, the real part shows a decreasing trend and tends to 1, and the imaginary part shows an increasing trend and tends to 0. The effective impedance matches the free-space impedance, and the wave-absorbing performance of the metamaterial device rises.

How is the hight, h1 and h2 determined? Why do you make the slot of VO2 circle? How the graphen pattern designed?

Response

Regarding the determination of h1 and h2, parameter optimisation was carried out by adjusting the structure through simulation. h1 is the value when the absorption bandwidth is at its maximum while ensuring that the absorptivity exceeds 90% in the operating range of Mode 1. While h2 is the value when the average transmissivity of the metamaterial in Mode 3 is at its maximum while ensuring that the Mode 1 and Mode 2 functionality is at its maximum.

The graphical design of vanadium dioxide has been addressed in the results and discussion section of the article. As can be seen in Fig. 2, the design process of the vanadium oxide patterned layer goes from three concentric rings and then to open rings. By opening the rings, an open resonance ring is formed to improve the absorption of the structure. Regarding the design of the graphene patterned layer, it is from a simple cross structure, and the structure is continuously optimised through simulation to achieve the function that the structure has an absorptivity exceeding 90% in the low frequency range of 2-5 THz.

Simulation details are missing. What is the simulator used? Please show the direction of incident wave and definition of polarization for TM and TE simulation. Comparison of TM and TE in figure 4b,c,d may be just because the polarization synmetry.

Response

In this paper, the CST simulation software is used, and specific additions regarding the simulation setup are made in the Design Methodology section. TE wave refers to an electromagnetic wave in which the electric field vector is perpendicular to the propagation direction and one component of the magnetic field vector is parallel to the propagation direction; TM wave refers to an electromagnetic wave in which the magnetic field vector is perpendicular to the propagation direction and one component of the electric field vector is parallel to the propagation direction. In the discussion of the simulation results, we add the definition of the polarisation of TE, TM waves.

Update to the manuscript:

Design and method

Where ,. By adjusting the temperature, the conductivityof vanadium dioxide can be varied in the range of 200 S/m to 2×105 S/m. In the experiment, vanadium dioxide is in the insulating state at  and in the metallic state at .

We performed simulations using CST Microwave Studio, applying periodic boundary conditions along the X and Y axes while setting open boundary conditions for the Z-axis. The electromagnetic wave was incident normally on the metamaterial surface along the negative Z-direction. Using a single-unit cell approach, we obtained the S-parameters to accelerate computations while maintaining accuracy. From these parameters, we calculated the reflectivity (R) and transmissivity (T). Eventually, the absorptivity can be described by the following equation[32,33]:

Simulation results in figure 8 should be discussed together with the physical mechanism. Advantage of the current design should be discussed in comparison to other references.

Response

In the elaboration about the electric field distribution, we also combine it with impedance matching. The better the impedance matching is, the easier the energy of the incident wave is absorbed by the metamaterial; the stronger the electric field localisation is, the more significant the energy concentration is, and the higher the absorptivity of the metamaterial is. The simulation results are verified while further elucidating the physical mechanism of metamaterial work.

Regarding the advantages of the metamaterials proposed in this paper, we have compared them with some previous similar studies to fully illustrate the advantages that the design has.

Update to the manuscript:

Result and discussion

From the comparisons in Table 1, we can see that the metamaterial proposed in this paper can provide a larger absorption bandwidth of 11.98 THz while ensuring that the absorptivity exceeds 90% compared to similar previous studies[34-37]. In addition to this, the bottom of the metamaterial device proposed in this paper uses vanadium dioxide as the bottom material compared to previous metamaterial devices. Different from the previous use of gold as the bottom material, vanadium dioxide has the advantage of converting the properties of the metamaterial device so that it can switch between absorption and transmission.

High-frequency electromagnetic waves can excite more resonant Modes of different scales and forms in metamaterials, which makes the electric field form strong localisation in multiple regions, which is essential for realising the absorption of electromagnetic waves in a wide bandwidth, and expanding the working frequency band of the metamaterial through the absorption of electromagnetic waves of different frequencies in different regions. Combined with the results given in Fig. 6(a), it can be seen that near the above frequencies, the fundamental part is all very close to 1 and the imaginary part is very close to 0. The input impedance matches with the spatial impedance, and the metamaterials can achieve a high absorption rate. This is consistent with the results demonstrated in Fig. 7 that the electric field is locally strong at each frequency, with significant energy concentration and high absorption efficiency of the metamaterial.

[1] D. Li, S. He, L. Su, H. Du, Y. Tian, Z. Gao, B. Xie, G. Huang, Switchable and tunable terahertz metamaterial absorber based on graphene and vanadium dioxide, Optical Materials 147 (2024) 114655. https://doi.org/10.1016/j.optmat.2023.114655.

[2] J. Zhao, H. Yang, X. Shan, X. Mi, S. Ma, Y. Huang, Research on dual-controlled terahertz metamaterial broadband absorber based on vanadium dioxide and graphene, Optics Communications 545 (2023) 129701. https://doi.org/10.1016/j.optcom.2023.129701.

[3] S. Nie, H. Feng, X. Li, P. Sun, Y. Zhou, L. Su, L. Ran, Y. Gao, A broadband absorber with multiple tunable functions for terahertz band based on graphene and vanadium dioxide, Diamond and Related Materials 139 (2023) 110374. https://doi.org/10.1016/j.diamond.2023.110374.

[4] C. Song, J. Wang, B. Zhang, Z. Qu, H. Jing, J. Kang, J. Hao, J. Duan, Dual-band/ultra-broadband switchable terahertz metamaterial absorber based on vanadium dioxide and graphene, Optics Communications 530 (2023) 129027. https://doi.org/10.1016/j.optcom.2022.129027.

The relation between figure 7 and figure 4(a) should be explained, especially the feature at 3 THz. For optimization of parameter, quantitative measure can be used for discussion, such as average transmission, minimum transmission and bandwidth. Although experimantal evaluation is not addressed in the paper, you should discuss reliability of the simulation results.

Response

Regarding the explanation of the physical mechanism of metamaterial work using the impedance matching principle, we link the simulation results with the results of impedance matching on the basis of the original article, and verify the physical mechanism of metamaterial work using the available results. When the input impedance and spatial impedance are matched, the absorption property of the metamaterial rises. Thus the higher absorptivity is obtained for metamaterials in the band where the real part is close to 1 and the imaginary part is close to 0.

In the discussion of parameter optimisation, we follow the comments and add the parameter of average absorptivity for quantitative discussion.

Update to the manuscript:

Result and discussion

Finally, we present the process of parameter optimisation during the design, including the simulation results of optimising the period p, the thickness of the vanadium dioxide pattern layer t2, and the relevant parameters of the graphene pattern layer d, r4, and r5, as shown in Figs. 9 and 10. Fig. 9(a)(b) demonstrate the effect of different p on the absorptivity of Mode 1 and Mode 2, respectively. From the figure, it can be seen that the absorptivity of the metamaterial in the mid-frequency region of 6-9 THz decreases gradually with the gradual increase of p from 19μm in Mode 1, and the absorptivity in some frequency ranges does not exceed 90%, which affects the overall broadband absorption effect. And the absorption bandwidth of the metamaterial in mode 1 reaches the maximum value of 11.98 THz when p = 19μm(dark red line). The overall absorptivity of the metamaterial in the operating frequency range in mode 2 exceeds 90% only when p = 19μm and p = 20μm(orange line). And the absorption bandwidth is 9.73 THz with an average absorptivity of 95.07% when p = 19μm; the absorption bandwidth is 9.09 THz with an average absorptivity of 94.08% when p = 20μm. In summary, p = 19μm is chosen as the final parameter. Fig. 9(c)(d) show the effect of different t2 on the absorptivity for Mode 1 and Mode 2, respectively. It can be seen that although at t2 = 0.3μm (pale yellow line), the absorption of the metamaterial in most of the frequency range in Mode 1 is better than the other results, the absorptivity near 12 THz in Mode 2 does not exceed 90%, which affects the overall broadband absorption effect. Whereas, at t2 = 0.25μm (orange line), the absorptivity of the metamaterial in the overall frequency range of both Mode 1 and Mode 2 exceeds 90%. Moreover, the average absorptivity of the metamaterial in the operating frequency range reaches 95.07% when t2 = 0.25μm, which is better than the average absorptivity of 94.2% when t2 = 0.3μm. Based on the above, t2 = 0.25μm is chosen.

 Fig. 10(a) shows the effect of different r4 on the absorptivity of Mode 1. From the figure, it can be seen that the absorption bandwidth of the metamaterial is the longest when r4 =7μm (purple line), but its absorptivity decreases significantly near 3 THz. Only when r4 = 5μm, the overall absorptivity of the metamaterial in the operating frequency range exceeds 90% and the absorption bandwidth of 11.98 THz reaches the maximum. Therefore, r4 = 5μm is chosen. Fig. 10(b) demonstrates the effect of different d on the Mode 1 absorptivity. Only when d = 10μm(pale yellow line) and d = 11μm(green line), the overall absorptivity of the metamaterial exceeds 90% in the operating frequency range. The absorption bandwidth is 11.94 THz when d = 11μm and 11.98 THz when d = 10μm. So d = 10μm is chosen as the final parameter. Fig. 10(c) demonstrates the effect of different r5 on the absorptivity of Mode 1. Only when r5 = 2.5μm(orange line) and r5 = 3μm(pale yellow line), the overall absorptivity of the metamaterial exceeds 90% in the operating frequency range. The absorption bandwidth is 11.77 THz when r5 = 2.5μm and 11.98 THz when r5 = 3μm. So r5 = 3μm is selected.

Followings are additional comments: Line 80, the reference to [22] looks wrong. The long explanation of figure 9 can be shortened.

Response

We have corrected the errors in the description of reference [22], and the description of the wide-angle absorptivity of metamaterials has been simplified accordingly.

Update to the manuscript:

Introduction

Zhang et al.[22] designed a metamaterial absorber based on vanadium dioxide, which can switch between absorption of more than 99% between 0.3 THz and 1.2 THz and high transmission at 6.2 THz.Zhang et al.[22] designed a metamaterial structure based on vanadium dioxide, which can obtain high absorptivity at frequencies of 3.53 THz, 4.98 THz, 6.70 THz, and 8.36 THz, with corresponding absorptivity values of 99.85%, 99.74%, 99.79%, and 99.56%.

Result and discussion

Since terahertz waves can be incident from different angles in practical applications, it is essential for metamaterial devices to have good wide-angle and polarisation insensitivity. Fig. 8 shows the absorptivity of TE and TM waves by three Modes of the metamaterial at different incidence angles. From Fig. 8(a)(b), it can be seen that under the TE polarisation condition, the metamaterial can achieve exceeding 90% absorptivity in Mode 1 for the incident waves with about 2-14 THz incidence angles from 0° to 50°; Under TM polarization, the absorptivity of metamaterials to incident waves with incident angles of 0° to 70° in the same frequency range exceeds 90%. From Fig. 8(c)(d), it can be seen that the metamaterials can achieve exceeding 90% absorptivity of incident waves in the range of 4-13.5 THz for incident angles of 0° to 60° under both TE and TM polarisation conditions, in Mode 2. Fig. 8(e) demonstrates that under TE polarisation conditions, the metamaterial can achieve exceeding 80% transmissivity for incident waves with an incident angle of 0°-30°at about 0.1-1.5 THz, 6.5-8 THz, and 13-15 THz, in Mode 3. Fig. 8(f) demonstrates that the metamaterial can achieve exceeding 90% transmissivity for incident waves with incident angles of approximately 0°-70° at 0.1-15 THz under TM polarization conditions in Mode 3. It can be seen that the metamaterial exhibits good polarization insensitivity as well as wide-angle absorption, which enables it to adapt to changes in incident angle and polarization angle in practical applications.

Reviewer 2:

The article entitled “Broadband Three-Mode Tunable Metamaterials Based on Graphene and Vanadium Oxide” by Wen et al. presents a metamaterial design where the absorption of terahertz radiation is tuned by the Fermi level of graphene or the conductivity of vanadium oxide. The topic is interesting and so is the suggested structure. Before considering the work for publication, I have some comments for the Authors:

Throughout the text there is very inconsistent way of writing equations and text. For example, there should always be a space between numerical values and units. Another example is “In the formulae,Ef,T,e,ω,ℏ,kBdenote” (no spaces). The Authors need to solve this with major revisions (text, figure and captions) before considering their work appropriate for publication.

Response

Regarding the formatting of formula letters as well as units, we have standardised the formatting of all formula letters.

Update to the manuscript:

Design and method

In the formulae,,,,,,,denote the Fermi energy level of graphene, ambient temperature, electron charge, carrier relaxation time, angular frequency of incident light, reduced Planck's constant and Boltzmann's constant, respectively. Since the frequency range investigated in this paper is within the terahertz band, the Fermi energy levels of graphene satisfy the relation:>>; and in this frequency range, the effect of interband jumps on the conductivity of graphene is negligible due to the bubbleley blocking effect[27]. In addition to this,>>. Based on all the conditions mentioned above, a simplified Drude Model can be used to describe the conductivity of graphene in the study of this paper[28] :

 In the equations,,,, and denote the Fermi energy level of graphene, the electron charge, the carrier relaxation time, the angular frequency of incident light, and the approximate Planck constant, respectively.

 From the equation, it can be seen that the electrical conductivity of graphene depends on the Fermi energy level of graphene , the carrier relaxation time  and the angular frequency of incident light  since the electronic charge and the reduced Planck's constant  are not affected by the experimental environment. Therefore, we can alter the Fermi energy level of graphene through chemical doping or applying a bias voltage to adjust the chemical potential, thereby modifying the conductivity of graphene. Among them, the adjustment range of graphene Fermi energy level is generally from 0 eV to 0.7 eV. In this paper, the carrier relaxation time  is selected.

The results in Figure 8 suggest that the mesh is poor. I suppose that the Authors run multiple calculations to optimize the structure, and one of them is what we see there. Is it so?

Response

Regarding the issue of substandard mesh quality, the maximum mesh quality following improvements reached 0.999, with an average mesh quality of 0.857. The mesh generation method selected is the surface-based default approach. This technique first meshes the model's surface before generating tetrahedral meshes internally. It effectively conforms to the geometric surface shape of the model, ensuring the accuracy of geometric features. Furthermore, the new electric field distribution diagram has been incorporated into the article.

Update to the manuscript:

Figure 7

There are mistakes and inconsistencies in the references.

Response

The problem of incorrect description in reference [22] has been corrected accordingly.

Update to the manuscript:

Introduction

Zhang et al.[22] designed a metamaterial absorber based on vanadium dioxide, which can switch between absorption of more than 99% between 0.3 THz and 1.2 THz and high transmission at 6.2 THz.Zhang et al.[22] designed a metamaterial structure based on vanadium dioxide, which can obtain high absorptivity at frequencies of 3.53 THz, 4.98 THz, 6.70 THz, and 8.36 THz, with corresponding absorptivity values of 99.85%, 99.74%, 99.79%, and 99.56%.

It would be helpful or even necessary to provide a scheme, maybe in Figure 1, of the model geometry. Some additional information in the text are also needed: package of finite element method calculations, mesh size, use of perfectly matched layers etc.

Response

In this paper, the CST simulation software is used, and specific additions regarding the simulation setup are made in the Design Methodology section. The specific settings are as follows:By setting periodic boundary conditions in the X and Y directions and open boundary conditions in the Z direction, the electromagnetic wave was incident perpendicular to the metamaterial device surface along the negative Z-axis. The S-parameters are obtained by using a single-cycle unit to simulate the whole cycle, which both speeds up the calculation and ensures the accuracy of the results.

The tunability of VO2 should be based on temperature-induced phase changes. How can this affect graphene?

Response

Regarding the issue of the effect of temperature variation on graphene, it is worth noting that during the temperature control of the vanadium dioxide dielectric constant, the temperature affects the concentration of graphene carriers, which indirectly affects the electrical conductivity of graphene. However, in the terahertz band, the temperature control range of vanadium dioxide is only within 100 °C, and the conductivity of graphene is much less affected by temperature than that of other bands, so that the influence of the weak conductivity change on the absorption characteristics of the metamaterial device is negligible. Therefore, the conductivity of graphene can be assumed to remain unchanged during the process of adjusting the vanadium dioxide dielectric constant using temperature.

How did you obtain the data shown in Figure 2?

Response

Manuscript ID: nanomaterials-3875084

Title: Broadband Three-Mode Tunable Metamaterials Based on Graphene and Vanadium Oxide

Corresponding Author: Zheng Qin, Tianyu Gao

Dear editors,

We would like to thank the reviewers for their thorough and careful reading of our manuscript, and  suggestions to improve our manuscript. We have carefully modified our manuscript strictly according to their suggestions, and hope that this revised manuscript can be accepted for publication.

We address in detail each point raised by the referee individually below; verbatim reviewer comment is

shown in blue, with our response in black under each comment. The corresponding changes in the

revised manuscript are marked in red for the deleted and highlight in yellow for the added content.

Sincerely yours,

Zheng Qin

***********************************************************************************

Response to the comments of the reviewers

***********************************************************************************

Reviewer 1:

The paper present the desing and simulation of metamaterial abrsorber in terahertz range using graphen and vabadium oxide. My major comment on the paper is the missing explanations on the desing principle, physical mechanism, simulation details and evaluation of the results. What is the design principle presented in figure 1? Please explain their physical mechanism or show appropriate reference explaining the physical mechanism. How is the bandwidth determined?

Response

We have supplemented the design principle, physical mechanism and simulation details accordingly. Especially in the physical mechanism, we combine the electric field distribution, impedance matching and simulation results to fully verify the physical mechanism of metamaterial operation and also the simulation results. Regarding the simulation details, we also give the relevant settings for simulation in the design methodology.

Update to the manuscript:

Result and discussion

We have investigated impedance matching and given the real and imaginary parts of the impedance matching of the metamaterial in the Mode 1 and Mode 2 operating bands, as shown in Fig. 6. Impedance matching requires that the relative impedance of the material tends to be close to 1 in the real part and 0 in the imaginary part; this is conducive to the reduction of reflection so that the energy can be better transmitted or absorbed. The imaginary part reflects the energy storage properties of the material; the smaller the value of the imaginary part, the weaker the resistance of the material will be, and the characteristics of the material will be closer to a pure resistance, which is consistent with the requirement of impedance matching for the imaginary part[39,40].

From Fig. 6(a), it can be seen that the real part (black line) do not converge to 1 (upper dashed line) and the imaginary part (red line) do not converge to 0 (lower dashed line) in the range of 0.1-2 THz. This indicates that the relative input impedance does not match with the free-space impedance, which makes the terahertz wave incident to the metamaterial device reflected at the interface, and the metamaterial device's absorption performance is poor, which is consistent with the simulation results obtained in Fig. 3(b).After 2 THz, the real part shows a rising trend and tends to 1, and the imaginary part shows a decreasing trend and tends to 0, and the effective impedance of the metamaterial matches with the free-space impedance, which achieves an effective absorption in the terahertz band. As can be seen in Fig. 6(b), near 3 THz, the real part is much larger than the imaginary part, indicating that the real part of the input impedance deviates from the corresponding real part of the free-space impedance, which results in strong reflection of the incident terahertz wave on the surface of the metamaterial device, and the absorption performance of the metamaterial device decreases, which corresponds to that of the absorption near 3 THz shown in Fig. 3(c). After 3 THz, the real part shows a decreasing trend and tends to 1, and the imaginary part shows an increasing trend and tends to 0. The effective impedance matches the free-space impedance, and the wave-absorbing performance of the metamaterial device rises.

How is the hight, h1 and h2 determined? Why do you make the slot of VO2 circle? How the graphen pattern designed?

Response

Regarding the determination of h1 and h2, parameter optimisation was carried out by adjusting the structure through simulation. h1 is the value when the absorption bandwidth is at its maximum while ensuring that the absorptivity exceeds 90% in the operating range of Mode 1. While h2 is the value when the average transmissivity of the metamaterial in Mode 3 is at its maximum while ensuring that the Mode 1 and Mode 2 functionality is at its maximum.

The graphical design of vanadium dioxide has been addressed in the results and discussion section of the article. As can be seen in Fig. 2, the design process of the vanadium oxide patterned layer goes from three concentric rings and then to open rings. By opening the rings, an open resonance ring is formed to improve the absorption of the structure. Regarding the design of the graphene patterned layer, it is from a simple cross structure, and the structure is continuously optimised through simulation to achieve the function that the structure has an absorptivity exceeding 90% in the low frequency range of 2-5 THz.

Simulation details are missing. What is the simulator used? Please show the direction of incident wave and definition of polarization for TM and TE simulation. Comparison of TM and TE in figure 4b,c,d may be just because the polarization synmetry.

Response

In this paper, the CST simulation software is used, and specific additions regarding the simulation setup are made in the Design Methodology section. TE wave refers to an electromagnetic wave in which the electric field vector is perpendicular to the propagation direction and one component of the magnetic field vector is parallel to the propagation direction; TM wave refers to an electromagnetic wave in which the magnetic field vector is perpendicular to the propagation direction and one component of the electric field vector is parallel to the propagation direction. In the discussion of the simulation results, we add the definition of the polarisation of TE, TM waves.

Update to the manuscript:

Design and method

Where ,. By adjusting the temperature, the conductivityof vanadium dioxide can be varied in the range of 200 S/m to 2×105 S/m. In the experiment, vanadium dioxide is in the insulating state at  and in the metallic state at .

We performed simulations using CST Microwave Studio, applying periodic boundary conditions along the X and Y axes while setting open boundary conditions for the Z-axis. The electromagnetic wave was incident normally on the metamaterial surface along the negative Z-direction. Using a single-unit cell approach, we obtained the S-parameters to accelerate computations while maintaining accuracy. From these parameters, we calculated the reflectivity (R) and transmissivity (T). Eventually, the absorptivity can be described by the following equation[32,33]:

Simulation results in figure 8 should be discussed together with the physical mechanism. Advantage of the current design should be discussed in comparison to other references.

Response

In the elaboration about the electric field distribution, we also combine it with impedance matching. The better the impedance matching is, the easier the energy of the incident wave is absorbed by the metamaterial; the stronger the electric field localisation is, the more significant the energy concentration is, and the higher the absorptivity of the metamaterial is. The simulation results are verified while further elucidating the physical mechanism of metamaterial work.

Regarding the advantages of the metamaterials proposed in this paper, we have compared them with some previous similar studies to fully illustrate the advantages that the design has.

Update to the manuscript:

Result and discussion

From the comparisons in Table 1, we can see that the metamaterial proposed in this paper can provide a larger absorption bandwidth of 11.98 THz while ensuring that the absorptivity exceeds 90% compared to similar previous studies[34-37]. In addition to this, the bottom of the metamaterial device proposed in this paper uses vanadium dioxide as the bottom material compared to previous metamaterial devices. Different from the previous use of gold as the bottom material, vanadium dioxide has the advantage of converting the properties of the metamaterial device so that it can switch between absorption and transmission.

High-frequency electromagnetic waves can excite more resonant Modes of different scales and forms in metamaterials, which makes the electric field form strong localisation in multiple regions, which is essential for realising the absorption of electromagnetic waves in a wide bandwidth, and expanding the working frequency band of the metamaterial through the absorption of electromagnetic waves of different frequencies in different regions. Combined with the results given in Fig. 6(a), it can be seen that near the above frequencies, the fundamental part is all very close to 1 and the imaginary part is very close to 0. The input impedance matches with the spatial impedance, and the metamaterials can achieve a high absorption rate. This is consistent with the results demonstrated in Fig. 7 that the electric field is locally strong at each frequency, with significant energy concentration and high absorption efficiency of the metamaterial.

[1] D. Li, S. He, L. Su, H. Du, Y. Tian, Z. Gao, B. Xie, G. Huang, Switchable and tunable terahertz metamaterial absorber based on graphene and vanadium dioxide, Optical Materials 147 (2024) 114655. https://doi.org/10.1016/j.optmat.2023.114655.

[2] J. Zhao, H. Yang, X. Shan, X. Mi, S. Ma, Y. Huang, Research on dual-controlled terahertz metamaterial broadband absorber based on vanadium dioxide and graphene, Optics Communications 545 (2023) 129701. https://doi.org/10.1016/j.optcom.2023.129701.

[3] S. Nie, H. Feng, X. Li, P. Sun, Y. Zhou, L. Su, L. Ran, Y. Gao, A broadband absorber with multiple tunable functions for terahertz band based on graphene and vanadium dioxide, Diamond and Related Materials 139 (2023) 110374. https://doi.org/10.1016/j.diamond.2023.110374.

[4] C. Song, J. Wang, B. Zhang, Z. Qu, H. Jing, J. Kang, J. Hao, J. Duan, Dual-band/ultra-broadband switchable terahertz metamaterial absorber based on vanadium dioxide and graphene, Optics Communications 530 (2023) 129027. https://doi.org/10.1016/j.optcom.2022.129027.

The relation between figure 7 and figure 4(a) should be explained, especially the feature at 3 THz. For optimization of parameter, quantitative measure can be used for discussion, such as average transmission, minimum transmission and bandwidth. Although experimantal evaluation is not addressed in the paper, you should discuss reliability of the simulation results.

Response

Regarding the explanation of the physical mechanism of metamaterial work using the impedance matching principle, we link the simulation results with the results of impedance matching on the basis of the original article, and verify the physical mechanism of metamaterial work using the available results. When the input impedance and spatial impedance are matched, the absorption property of the metamaterial rises. Thus the higher absorptivity is obtained for metamaterials in the band where the real part is close to 1 and the imaginary part is close to 0.

In the discussion of parameter optimisation, we follow the comments and add the parameter of average absorptivity for quantitative discussion.

Update to the manuscript:

Result and discussion

Finally, we present the process of parameter optimisation during the design, including the simulation results of optimising the period p, the thickness of the vanadium dioxide pattern layer t2, and the relevant parameters of the graphene pattern layer d, r4, and r5, as shown in Figs. 9 and 10. Fig. 9(a)(b) demonstrate the effect of different p on the absorptivity of Mode 1 and Mode 2, respectively. From the figure, it can be seen that the absorptivity of the metamaterial in the mid-frequency region of 6-9 THz decreases gradually with the gradual increase of p from 19μm in Mode 1, and the absorptivity in some frequency ranges does not exceed 90%, which affects the overall broadband absorption effect. And the absorption bandwidth of the metamaterial in mode 1 reaches the maximum value of 11.98 THz when p = 19μm(dark red line). The overall absorptivity of the metamaterial in the operating frequency range in mode 2 exceeds 90% only when p = 19μm and p = 20μm(orange line). And the absorption bandwidth is 9.73 THz with an average absorptivity of 95.07% when p = 19μm; the absorption bandwidth is 9.09 THz with an average absorptivity of 94.08% when p = 20μm. In summary, p = 19μm is chosen as the final parameter. Fig. 9(c)(d) show the effect of different t2 on the absorptivity for Mode 1 and Mode 2, respectively. It can be seen that although at t2 = 0.3μm (pale yellow line), the absorption of the metamaterial in most of the frequency range in Mode 1 is better than the other results, the absorptivity near 12 THz in Mode 2 does not exceed 90%, which affects the overall broadband absorption effect. Whereas, at t2 = 0.25μm (orange line), the absorptivity of the metamaterial in the overall frequency range of both Mode 1 and Mode 2 exceeds 90%. Moreover, the average absorptivity of the metamaterial in the operating frequency range reaches 95.07% when t2 = 0.25μm, which is better than the average absorptivity of 94.2% when t2 = 0.3μm. Based on the above, t2 = 0.25μm is chosen.

 Fig. 10(a) shows the effect of different r4 on the absorptivity of Mode 1. From the figure, it can be seen that the absorption bandwidth of the metamaterial is the longest when r4 =7μm (purple line), but its absorptivity decreases significantly near 3 THz. Only when r4 = 5μm, the overall absorptivity of the metamaterial in the operating frequency range exceeds 90% and the absorption bandwidth of 11.98 THz reaches the maximum. Therefore, r4 = 5μm is chosen. Fig. 10(b) demonstrates the effect of different d on the Mode 1 absorptivity. Only when d = 10μm(pale yellow line) and d = 11μm(green line), the overall absorptivity of the metamaterial exceeds 90% in the operating frequency range. The absorption bandwidth is 11.94 THz when d = 11μm and 11.98 THz when d = 10μm. So d = 10μm is chosen as the final parameter. Fig. 10(c) demonstrates the effect of different r5 on the absorptivity of Mode 1. Only when r5 = 2.5μm(orange line) and r5 = 3μm(pale yellow line), the overall absorptivity of the metamaterial exceeds 90% in the operating frequency range. The absorption bandwidth is 11.77 THz when r5 = 2.5μm and 11.98 THz when r5 = 3μm. So r5 = 3μm is selected.

Followings are additional comments: Line 80, the reference to [22] looks wrong. The long explanation of figure 9 can be shortened.

Response

We have corrected the errors in the description of reference [22], and the description of the wide-angle absorptivity of metamaterials has been simplified accordingly.

Update to the manuscript:

Introduction

Zhang et al.[22] designed a metamaterial absorber based on vanadium dioxide, which can switch between absorption of more than 99% between 0.3 THz and 1.2 THz and high transmission at 6.2 THz.Zhang et al.[22] designed a metamaterial structure based on vanadium dioxide, which can obtain high absorptivity at frequencies of 3.53 THz, 4.98 THz, 6.70 THz, and 8.36 THz, with corresponding absorptivity values of 99.85%, 99.74%, 99.79%, and 99.56%.

Result and discussion

Since terahertz waves can be incident from different angles in practical applications, it is essential for metamaterial devices to have good wide-angle and polarisation insensitivity. Fig. 8 shows the absorptivity of TE and TM waves by three Modes of the metamaterial at different incidence angles. From Fig. 8(a)(b), it can be seen that under the TE polarisation condition, the metamaterial can achieve exceeding 90% absorptivity in Mode 1 for the incident waves with about 2-14 THz incidence angles from 0° to 50°; Under TM polarization, the absorptivity of metamaterials to incident waves with incident angles of 0° to 70° in the same frequency range exceeds 90%. From Fig. 8(c)(d), it can be seen that the metamaterials can achieve exceeding 90% absorptivity of incident waves in the range of 4-13.5 THz for incident angles of 0° to 60° under both TE and TM polarisation conditions, in Mode 2. Fig. 8(e) demonstrates that under TE polarisation conditions, the metamaterial can achieve exceeding 80% transmissivity for incident waves with an incident angle of 0°-30°at about 0.1-1.5 THz, 6.5-8 THz, and 13-15 THz, in Mode 3. Fig. 8(f) demonstrates that the metamaterial can achieve exceeding 90% transmissivity for incident waves with incident angles of approximately 0°-70° at 0.1-15 THz under TM polarization conditions in Mode 3. It can be seen that the metamaterial exhibits good polarization insensitivity as well as wide-angle absorption, which enables it to adapt to changes in incident angle and polarization angle in practical applications.

Reviewer 2:

The article entitled “Broadband Three-Mode Tunable Metamaterials Based on Graphene and Vanadium Oxide” by Wen et al. presents a metamaterial design where the absorption of terahertz radiation is tuned by the Fermi level of graphene or the conductivity of vanadium oxide. The topic is interesting and so is the suggested structure. Before considering the work for publication, I have some comments for the Authors:

Throughout the text there is very inconsistent way of writing equations and text. For example, there should always be a space between numerical values and units. Another example is “In the formulae,Ef,T,e,ω,ℏ,kBdenote” (no spaces). The Authors need to solve this with major revisions (text, figure and captions) before considering their work appropriate for publication.

Response

Regarding the formatting of formula letters as well as units, we have standardised the formatting of all formula letters.

Update to the manuscript:

Design and method

In the formulae,,,,,,,denote the Fermi energy level of graphene, ambient temperature, electron charge, carrier relaxation time, angular frequency of incident light, reduced Planck's constant and Boltzmann's constant, respectively. Since the frequency range investigated in this paper is within the terahertz band, the Fermi energy levels of graphene satisfy the relation:>>; and in this frequency range, the effect of interband jumps on the conductivity of graphene is negligible due to the bubbleley blocking effect[27]. In addition to this,>>. Based on all the conditions mentioned above, a simplified Drude Model can be used to describe the conductivity of graphene in the study of this paper[28] :

 In the equations,,,, and denote the Fermi energy level of graphene, the electron charge, the carrier relaxation time, the angular frequency of incident light, and the approximate Planck constant, respectively.

 From the equation, it can be seen that the electrical conductivity of graphene depends on the Fermi energy level of graphene , the carrier relaxation time  and the angular frequency of incident light  since the electronic charge and the reduced Planck's constant  are not affected by the experimental environment. Therefore, we can alter the Fermi energy level of graphene through chemical doping or applying a bias voltage to adjust the chemical potential, thereby modifying the conductivity of graphene. Among them, the adjustment range of graphene Fermi energy level is generally from 0 eV to 0.7 eV. In this paper, the carrier relaxation time  is selected.

The results in Figure 8 suggest that the mesh is poor. I suppose that the Authors run multiple calculations to optimize the structure, and one of them is what we see there. Is it so?

Response

Regarding the issue of substandard mesh quality, the maximum mesh quality following improvements reached 0.999, with an average mesh quality of 0.857. The mesh generation method selected is the surface-based default approach. This technique first meshes the model's surface before generating tetrahedral meshes internally. It effectively conforms to the geometric surface shape of the model, ensuring the accuracy of geometric features. Furthermore, the new electric field distribution diagram has been incorporated into the article.

Update to the manuscript:

Figure 7

There are mistakes and inconsistencies in the references.

Response

The problem of incorrect description in reference [22] has been corrected accordingly.

Update to the manuscript:

Introduction

Zhang et al.[22] designed a metamaterial absorber based on vanadium dioxide, which can switch between absorption of more than 99% between 0.3 THz and 1.2 THz and high transmission at 6.2 THz.Zhang et al.[22] designed a metamaterial structure based on vanadium dioxide, which can obtain high absorptivity at frequencies of 3.53 THz, 4.98 THz, 6.70 THz, and 8.36 THz, with corresponding absorptivity values of 99.85%, 99.74%, 99.79%, and 99.56%.

It would be helpful or even necessary to provide a scheme, maybe in Figure 1, of the model geometry. Some additional information in the text are also needed: package of finite element method calculations, mesh size, use of perfectly matched layers etc.

Response

In this paper, the CST simulation software is used, and specific additions regarding the simulation setup are made in the Design Methodology section. The specific settings are as follows:By setting periodic boundary conditions in the X and Y directions and open boundary conditions in the Z direction, the electromagnetic wave was incident perpendicular to the metamaterial device surface along the negative Z-axis. The S-parameters are obtained by using a single-cycle unit to simulate the whole cycle, which both speeds up the calculation and ensures the accuracy of the results.

The tunability of VO2 should be based on temperature-induced phase changes. How can this affect graphene?

Response

Regarding the issue of the effect of temperature variation on graphene, it is worth noting that during the temperature control of the vanadium dioxide dielectric constant, the temperature affects the concentration of graphene carriers, which indirectly affects the electrical conductivity of graphene. However, in the terahertz band, the temperature control range of vanadium dioxide is only within 100 °C, and the conductivity of graphene is much less affected by temperature than that of other bands, so that the influence of the weak conductivity change on the absorption characteristics of the metamaterial device is negligible. Therefore, the conductivity of graphene can be assumed to remain unchanged during the process of adjusting the vanadium dioxide dielectric constant using temperature.

How did you obtain the data shown in Figure 2?

Response

Reviewer 2 Report

Comments and Suggestions for Authors

The article entitled “Broadband Three-Mode Tunable Metamaterials Based on Graphene and Vanadium Oxide” by Wen et al. presents a metamaterial design where the absorption of terahertz radiation is tuned by the Fermi level of graphene or the conductivity of vanadium oxide. The topic is interesting and so is the suggested structure. Before considering the work for publication, I have some comments for the Authors:

  • Throughout the text there is very inconsistent way of writing equations and text. For example, there should always be a space between numerical values and units. Another example is “In the formulae,Ef,T,e,𝜏,ω,ℏ,kBdenote” (no spaces). The Authors need to solve this with major revisions (text, figure and captions) before considering their work appropriate for publication.
  • The results in Figure 8 suggest that the mesh is poor. I suppose that the Authors run multiple calculations to optimize the structure, and one of them is what we see there. Is it so?
  • There are mistakes and inconsistencies in the references.
  • It would be helpful or even necessary to provide a scheme, maybe in Figure 1, of the model geometry. Some additional information in the text are also needed: package of finite element method calculations, mesh size, use of perfectly matched layers etc.
  • The tunability of VO2 should be based on temperature-induced phase changes. How can this affect graphene?
  • How did you obtain the data shown in Figure 2?

Author Response

The article entitled “Broadband Three-Mode Tunable Metamaterials Based on Graphene and Vanadium Oxide” by Wen et al. presents a metamaterial design where the absorption of terahertz radiation is tuned by the Fermi level of graphene or the conductivity of vanadium oxide. The topic is interesting and so is the suggested structure. Before considering the work for publication, I have some comments for the Authors:

Throughout the text there is very inconsistent way of writing equations and text. For example, there should always be a space between numerical values and units. Another example is “In the formulae,Ef,T,e,ω,ℏ,kBdenote” (no spaces). The Authors need to solve this with major revisions (text, figure and captions) before considering their work appropriate for publication.

Response

Regarding the formatting of formula letters as well as units, we have standardised the formatting of all formula letters.

The results in Figure 8 suggest that the mesh is poor. I suppose that the Authors run multiple calculations to optimize the structure, and one of them is what we see there. Is it so?

Response

Regarding the issue of substandard mesh quality, the maximum mesh quality following improvements reached 0.999, with an average mesh quality of 0.857. The mesh generation method selected is the surface-based default approach. This technique first meshes the model's surface before generating tetrahedral meshes internally. It effectively conforms to the geometric surface shape of the model, ensuring the accuracy of geometric features. Furthermore, the new electric field distribution diagram has been incorporated into the article.

Update to the manuscript:

Figure 7

There are mistakes and inconsistencies in the references.

Response

The problem of incorrect description in reference [22] has been corrected accordingly.

Update to the manuscript:

Introduction

Zhang et al.[22] designed a metamaterial absorber based on vanadium dioxide, which can switch between absorption of more than 99% between 0.3 THz and 1.2 THz and high transmission at 6.2 THz.Zhang et al.[22] designed a metamaterial structure based on vanadium dioxide, which can obtain high absorptivity at frequencies of 3.53 THz, 4.98 THz, 6.70 THz, and 8.36 THz, with corresponding absorptivity values of 99.85%, 99.74%, 99.79%, and 99.56%.

It would be helpful or even necessary to provide a scheme, maybe in Figure 1, of the model geometry. Some additional information in the text are also needed: package of finite element method calculations, mesh size, use of perfectly matched layers etc.

Response

In this paper, the CST simulation software is used, and specific additions regarding the simulation setup are made in the Design Methodology section. The specific settings are as follows:By setting periodic boundary conditions in the X and Y directions and open boundary conditions in the Z direction, the electromagnetic wave was incident perpendicular to the metamaterial device surface along the negative Z-axis. The S-parameters are obtained by using a single-cycle unit to simulate the whole cycle, which both speeds up the calculation and ensures the accuracy of the results.

The tunability of VO2 should be based on temperature-induced phase changes. How can this affect graphene?

Response

Regarding the issue of the effect of temperature variation on graphene, it is worth noting that during the temperature control of the vanadium dioxide dielectric constant, the temperature affects the concentration of graphene carriers, which indirectly affects the electrical conductivity of graphene. However, in the terahertz band, the temperature control range of vanadium dioxide is only within 100 °C, and the conductivity of graphene is much less affected by temperature than that of other bands, so that the influence of the weak conductivity change on the absorption characteristics of the metamaterial device is negligible. Therefore, the conductivity of graphene can be assumed to remain unchanged during the process of adjusting the vanadium dioxide dielectric constant using temperature.

How did you obtain the data shown in Figure 2?

Response

Round 2

Reviewer 1 Report

Comments and Suggestions for Authors

I appreciated the revision of paper, however your achievement is not clear and the content include many mistakes. Some contents are not physically understandable. Reference papers are wrongly addressed. Abstract is missing the word "Terahertz" on the first line. All these points should have been workded out before submission.

References should be properly addressed. Ref[12] (Wen et al. ?)Ref [19] (Yang et al. ?) and Ref[22] (Ge et al. ?) are wrongly addressed. 

Equation numbering is wrongly indicated. Most of explanation after equation (4) can be deleted or shortened, because of multiple definition or too simple an explanation.

Ge et al. [22] presented absorption–transmission switchable devices using graphen and VO2. They also discussed polarization and angle of incidence. These should be addressed in the paper and included in table 1.

Required performance of the metamaterial should be clearly explained, though you often write more than 90% absorptivity.  How about bandwidth requirements? What is the targeted performance for parameter optimization ?

The desing principle should be explained into details. Why the multiple ring structure ? Why the cut in the ring improves performance ? How the ring size is designed in relation to the bandwidth ?

For graphen, only the simulated pattern is shown, but the structure is a combination of crosses and circles. Why you use these combination ? You should address their physical mechanism in the explanation.

To my knowledge, the definition of TE or TM is for waveguide mode and not appropriate for input electromagnetic radiation as plane wave, and the definition of polarization look completely wrong. I undestand that the polarization insensitivity is just because the patterns is polarization synmetrical. Also, refer Ge et al. [22].

A word "waveguide" is wrongly used in page 3.

Instead of equation (8), I recommend to include an equation to relate relative impedance and absorptivity for better understanding.

To summarize, what is new in the design and what is new achievements ? These should be explained clearly with the physical mechanism. 

Author Response

I appreciated the revision of paper, however your achievement is not clear and the content include many mistakes. Some contents are not physically understandable. Reference papers are wrongly addressed. Abstract is missing the word "Terahertz" on the first line. All these points should have been workded out before submission.

References should be properly addressed. Ref[12] (Wen et al. ?)Ref [19] (Yang et al. ?) and Ref[22] (Ge et al. ?) are wrongly addressed.

Response:

We've corrected the error and re-checked to make sure the reference is correct.

Update to the manuscript:

In 2009, Li Wen et al.[12] designed a terahertz metamaterial structure that achieved exceeding 90% absorptivity from 0.8 THz to 1.2 THz.

Li Yang et al.[19] utilize graphene to design a metamaterial structure, achieving an absorptivity exceeding 99% for the two peaks at 3.85 THz and 5.04 THz.

Zhang et al.[22] designed a metamaterial structure based on , which can obtain high absorptivity at frequencies of 3.53 THz, 4.98 THz, 6.70 THz, and 8.36 THz, with corresponding absorptivity values of 99.85%, 99.74%, 99.79%, and 99.56%. Ge et al.[22] designed a metamaterial structure which can switch between four narrow-band absorption peaks and high transmissivity transmission. In the transmission mode, 98.2% transmissivity is achieved at 6.2 THz and the reflectivity does not exceed 3% over the operating frequency range.

Equation numbering is wrongly indicated. Most of explanation after equation (4) can be deleted or shortened, because of multiple definition or too simple an explanation.

Response:

As suggested, we simplify the description of equation (4).

Update to the manuscript:

From the equation, it can be seen that the electrical conductivity of graphene  depends on the Fermi energy level of graphene , the carrier relaxation time  and the angular frequency of incident light  since the electronic charge and the reduced Planck's constant  are not affected by the experimental environment. Therefore, we can alter the Fermi energy level of graphene through chemical doping or applying a bias voltage to adjust the chemical potential, thereby modifying the conductivity of graphene . Among them, the adjustment range of graphene Fermi energy level is generally from 0 eV to 0.9 eV. In this paper, the carrier relaxation time  is selected.

Ge et al. [22] presented absorption–transmission switchable devices using graphen and VO2. They also discussed polarization and angle of incidence. These should be addressed in the paper and included in table 1.

Response:

Since the study in Ref. [22] achieved multimodal absorption when VO2 is in its metallic state, it is not possible to directly compare it to the broadband absorption in this paper. In addition to this, reference [22] does not specify the average transmittance, but only mentions the highest transmittance and the corresponding frequency, which makes it impossible to compare by mean transmittance in the table. To this end, we introduced more appropriate references [38] and [39] and compared them with the studies in this paper.

Update to the manuscript:

From the comparisons in Table 1, we can see that the metamaterial proposed in this paper can provide a larger absorption bandwidth of 11.98 THz while ensuring that the absorptivity exceeds 90% compared to similar previous studies[34-37]. In addition to this, the bottom of the metamaterial device proposed in this paper uses VO2 as the bottom material compared to previous metamaterial devices. Different from the previous use of gold as the bottom material, VO2 has the advantage of converting the properties of the metamaterial device so that it can switch between absorption and transmission.

As can be seen from the comparison in Table 1, the advantage of the metamaterial proposed in this paper over previous broadband absorption studies is that it can obtain two switchable ultra-broad absorption bands of 11.98 THz and 9.73 THz while guaranteeing an absorptivity exceeding 90% [34-39]. Although the switching function between absorption and transmission has been investigated using VO2 materials (e.g., reference [22] achieved switching between multiband absorption and high transmissivity transmission; references [38] and [39] achieved switching between broadband absorption and transmission), there is a lack of research on switching between ultra-broadband absorption and high-transmissivity modes. The proposed metamaterial structure can realize the mode switching between two ultra-wideband absorption and the function switching between ultra-wideband absorption and broadband transmission, with an average transmissivity of 88% in the operating frequency range, which effectively fills this research gap.

References

Function(absorption band)

Broad bandwidth >90% (THz)

Performance

Average transmissivity

34

Broadband &

Multiband (three peaks)

1.56 (3.21-4.77)

Only absorption

_

35

Broadband

5.02 (1.11-6.13)

Only absorption

_

36

Broadband

3.22 (3.74-6.96)

Only absorption

_

37

Broadband &

Multiband (two peaks)

3.37 (0.94-4.31)

Only absorption

_

38

Broadband

5.80 (3.60-9.40)

Absorption and transmission

86% (5.3-10THz)

39

Broadband

4.39 (2.46-6.85)

Absorption and transmission

69.61% (0.01-2.32THz)

69.79%(6.82-10.00THz)

This work

Dual ultra-wideband

11.98 (2.05-14.03 )

9.73 (4.07-13.80)

Absorption and transmission

88% (0.1-15THz)

[34] D. Li, S. He, L. Su, H. Du, Y. Tian, Z. Gao, B. Xie, G. Huang, Switchable and tunable terahertz metamaterial absorber based on graphene and vanadium dioxide, Optical Materials 147 (2024) 114655. https://doi.org/10.1016/j.optmat.2023.114655.

[35] J. Zhao, H. Yang, X. Shan, X. Mi, S. Ma, Y. Huang, Research on dual-controlled terahertz metamaterial broadband absorber based on vanadium dioxide and graphene, Optics Communications 545 (2023) 129701. https://doi.org/10.1016/j.optcom.2023.129701.

[36] S. Nie, H. Feng, X. Li, P. Sun, Y. Zhou, L. Su, L. Ran, Y. Gao, A broadband absorber with multiple tunable functions for terahertz band based on graphene and vanadium dioxide, Diamond and Related Materials 139 (2023) 110374. https://doi.org/10.1016/j.diamond.2023.110374.

[37] C. Song, J. Wang, B. Zhang, Z. Qu, H. Jing, J. Kang, J. Hao, J. Duan, Dual-band/ultra-broadband switchable terahertz metamaterial absorber based on vanadium dioxide and graphene, Optics Communications 530 (2023) 129027. https://doi.org/10.1016/j.optcom.2022.129027.

[38] Y. Zhang, B. Hou, Q. Song, Z. Yi, Q. Zeng, A THz smart switch based on a VO2 metamaterial that switches between wide-angle ultra-wideband absorption and transmission, Dalton Transactions 53 (2024) 19264–19271. 10.1039/d4dt02475c.

[39] W. Lu, W. Zhang, Q. Song, Z. Yi, S. Cheng, B. Tang, Q. Zeng, P. Wu, S. Ahmad, Terahertz smart devices based on phase change material VO2 and metamaterial graphene that combine thermally adjustable absorption and selective transmission, Optics & Laser Technology 181 (2025) 111928. https://doi.org/10.1016/j.optlastec.2024.111928.

Required performance of the metamaterial should be clearly explained, though you often write more than 90% absorptivity. How about bandwidth requirements? What is the targeted performance for parameter optimization?

Response:

We define the frequency band range with absorption rate greater than 90% as the absorption bandwidth. Our primary optimization goal is the absorption bandwidth when VO2 is in the metallic state, while the average transmittance of VO2 in the dielectric state is the secondary optimization goal.

Update to the manuscript:

We define the frequency band range with absorption rate greater than 90% as the absorption bandwidth, and select the absorption bandwidth of VO2 in the metallic state as the main optimization goal, and the average transmittance of VO2 in the dielectric state as the secondary optimization target. After several rounds of simulation optimisation, we obtain the optimal structural parameters: r1 = 1.5 μm, r2 = 4.5 μm, r3 = 7.5 μm, r4 = 5 μm, r5 = 3 μm, a = 2 μm, b = 1 μm, c = 1.5 μm, d = 10 μm, t1 = 0.3 μm, t2 = 0.25 μm, h1 = 4.25 μm, h2 = 7.7 μm, p = 19 μm.

The desing principle should be explained into details. Why the multiple ring structure ? Why the cut in the ring improves performance? How the ring size is designed in relation to the bandwidth?

Response:

The VO2 structure is designed as a concentric ring, and the core purpose is to solve the problems of narrow absorption bandwidth and high polarization sensitivity of the traditional metamaterial absorber through the coupling of multiple resonance modes and the optimisation of structural symmetry, and ultimately to achieve ultra-wideband, polarization-insensitive terahertz wave absorption performance[1]. The essence of absorbing terahertz waves is to dissipate or transform the electromagnetic energy through the coupling of the resonant structure with the incident electromagnetic wave. While a single size resonant structure can only form resonant absorption at a specific frequency, the concentric ring design builds three independent resonant units by having three rings of different radii: outer, middle, and inner. Each ring can generate resonant absorption at different frequencies in the terahertz band, and there is an electromagnetic field coupling effect between neighbouring rings, which can further broaden the coverage of the resonant frequency, and ultimately achieve the performance of ultra-wideband absorption. The direction of incident terahertz wave polarization affects the absorption effect of asymmetric structures. On the other hand, the concentric ring has rotational symmetry, and no matter how the polarization angle of the incident terahertz wave changes, the coupling between the structure and the electromagnetic wave remains the same, so the absorption intensity is basically unchanged. Simulation results also verify this property - the absorption bandwidth and absorption rate do not fluctuate significantly at different incident polarization angles, which meets the demand for polarization stability in practical applications.

Regarding why cutting the rings enhances the absorptivity, it is because the open resonant ring structure formed by cutting can excite stronger localized surface plasmon resonance, enhance the coupling between the electromagnetic field and the structure, and significantly reduce the reflectivity. Specifically, the four openings of the outer ring are equivalent to four parallel slit capacitors, and the structure of the middle ring also forms a similar capacitive effect in the same way. When terahertz electromagnetic waves are incident perpendicularly on this open resonant ring structure and the incident electric field component is parallel to the open slits, the entire structure can be equivalent to an LC oscillation circuit—the metal ring part induces eddy currents due to the alternating magnetic field, which is equivalent to inductance L; the slits at the openings form an equivalent capacitance C due to charge accumulation. When the frequency of the incident terahertz wave matches the intrinsic resonance frequency of this LC circuit, a strong resonance effect is triggered: at this time, the electric field at the slits is greatly enhanced (local field enhancement effect), and the eddy current loss in the metal ring is significantly increased, so that electromagnetic energy is efficiently absorbed through forms such as Joule heat, thereby exhibiting high absorptivity. We show the change in reflectivity of the metamaterial before and after ring cutting in Mode 2 through Fig. 1: The reflectivity when uncut (gray line) is significantly higher than that after cutting (blue line) in most frequency ranges, with the difference often exceeding 10%, which is consistent with the characteristic that the uncut closed ring structure is difficult to form effective LC resonance and electromagnetic energy is more reflected rather than absorbed, so its absorption performance is poor.

By using impedance matching theory, the geometric parameters of VO₂ concentric rings (including radius and width) were optimized to tune the absorber's effective permittivity (ε) and permeability (μ), enabling its impedance to approach that of free space. Simulation results show that for Mode 1 (2.05-14.03 THz) and Mode 2 (4.07-13.80 THz), the absorber exhibits impedance characteristics where the real part is close to 1 and the imaginary part is near 0, achieving near-perfect impedance matching with free space. This optimized impedance matching significantly reduces interface reflection losses, thereby ensuring high-efficiency ultra-broadband absorption performance across both operational modes.

Fig. 1. Reflectivity plots of uncut rings before and after cutting them

[1] G. Wu, X. Jiao, Y. Wang, Z. Zhao, Y. Wang, J. Liu, Ultra-wideband tunable metamaterial perfect absorber based on vanadium dioxide, Optics Express 29 (2021). 10.1364/oe.416227.

For graphene, only the simulated pattern is shown, but the structure is a combination of crosses and circles. Why you use these combination? You should address their physical mechanism in the explanation.

Response:

For the structure of the graphene layer, by simulating the absorption performance of the cross structure alone and the four-quarter-circle structure alone, it is found that there are obvious limitations of the single structure, as is shown in Fig. 2, and it is impossible to achieve the good broadband absorption effect of Mode 1 in the frequency band of 2-4 THz. It is because of the insufficiency of the single structure in the absorption bandwidth that a combination of cross and quarter-circle structures is chosen for the study. Each of the two structures corresponds to the absorptivity peaks at a specific frequency, and the combination of the two structures can realise the "peak complementarity", moreover, both the horizontal and vertical arms of the cross structure can form a local electric field superposition region with the curved edge of the quarter-circle structure to enhance the absorption of terahertz waves, breaking through the limitations of the single structure and realising the broadband absorption at a low frequency band of 2-4 THz. What's more, the combination of these two shapes arranges more resonators in a limited unit area compared to a single shape. This enables the structure to interact with electromagnetic waves in multiple ways within the terahertz frequency range, thus expanding the absorption bandwidth.

In addition, the main reason for not choosing a bar for the cross structure in the middle, but adding a half-circle on top of the bar, is that the bar structure is susceptible to the polarization direction, and the absorption intensity varies significantly with the polarization angle. The structure of the graphene layer has a combination of crosses and four quarter circles, which also has symmetry, so that neither the absorption bandwidth nor the absorptivity fluctuates significantly at different incident polarization angles.

Fig. 2. (gray line) Graphene patterned layers are only cross-shaped; (blue line) Graphene patterned layers are only circular; (red line) Graphene patterned layers are both circular and cross-shaped

To my knowledge, the definition of TE or TM is for waveguide mode and not appropriate for input electromagnetic radiation as plane wave, and the definition of polarization look completely wrong. I undestand that the polarization insensitivity is just because the patterns is polarization synmetrical. Also, refer Ge et al. [22].

Response:

The direction of propagation, the direction of the electric field and the direction of the magnetic field are perpendicular to each other. Our definition of TE waves and TM waves is: TM wave represents the linearly polarized light with electric field along the X-axis when the incident direction is the Z-axis; TE wave represents the linearly polarized light with electric field along the Y-axis when the incident direction is the Z-axis.

Update to the manuscript:

The absorption as well as transmission spectra of the three patterns of the metamaterial designed in this paper are shown in Fig. 3. TE waves are electromagnetic waves in which the electric field vector is perpendicular to the direction of propagation and a component of the magnetic field vector exists parallel to the direction of propagation; TM waves are electromagnetic waves in which the magnetic field vector is perpendicular to the direction of propagation and a component of the electric field vector exists parallel to the direction of propagation. TM wave represents the linearly polarized light with electric field along the X-axis when the incident direction is the Z-axis; TE wave represents the linearly polarized light with electric field along the Y-axis when the incident direction is the Z-axis. As can be seen from Fig. 3(b)(c)(d), the curves obtained from the simulation are highly consistent when the incident wave is a TE wave and a TM wave in the three Modes, respectively, indicating that the metamaterial is insensitive to the polarization angle of the incident light.

A word "waveguide" is wrongly used in page 3.

Response:

We've corrected that mistake.

Update to the manuscript:

The simulation results show that by controlling the conductivity of VO2 and the Fermi energy level of graphene, the absorbing waveguide the metamaterial can be switched between 2.05-14.03 THz long broadband absorption (Mode 1), 4.07-13.80 THz short broadband absorption (Mode 2) as well as transmission (Mode 3), and the absorptivity exceeds 90%, and the transmissivity exceeds 80% in all cases.

Instead of equation (8), I recommend to include an equation to relate relative impedance and absorptivity for better understanding.

Response:

The relationship regarding impedance and absorptivity can be expressed by the following equation:

Here  is the characteristic impedance, and the equation is based on the principle of energy conservation and the scattering parameter relationship in microwave network theory, which explicitly states the relationship between relative impedance and absorptivity.

However, the purpose of introducing equation (8) in the article is mainly to explain the physical mechanism of metamaterials working from the point of view of the impedance matching principle, rather than discussing the relationship between impedance matching and absorptivity. Therefore we think it would be better to keep equation (8).

To summarize, what is new in the design and what is new achievements ? These should be explained clearly with the physical mechanism.

Response:

The core advantage of the metamaterials proposed in this paper is the ability to switch between ultra-wideband absorption and high-transmissivity transmission. In broadband absorption mode, the bandwidth of absorption reaches 11.98 THz; in transmission mode, the average transmissivity in the operating frequency range reaches 88%. In addition to the comparisons in Table 1 above, we have added information about the innovativeness of the article in the conclusion section.

Update to the manuscript:

In this paper, we propose a terahertz metamaterial design based on graphene as well as VO2 materials. The structure consists of five layers: the bottom VO2 layer, the silicon dioxide dielectric layer, the VO2 patterned layer, the silicon dioxide dielectric layer, and the top graphene patterned layer. Through simulation, we conclude that it is can operate in three different Modes, which can achieve long broadband absorption in the frequency range of 2.05-14.03 THz with an absorptivity exceeding 90%, shorter broadband absorption in the frequency range of 4.07-13.80 THz with an absorptivity exceeding 90%, and high transmission in the frequency range of 0.1-15 THz with an average transmittance of 88%. Its core advantage lies in its ability to switch between ultra-broadband absorption and high-transmissivity transmission. To elucidate the physical mechanism of metamaterial operation, we have presented and analyzed the real and imaginary parts of the impedance matching of the metamaterial in the operating frequency bands of Mode 1 and Mode 2, based on the principle of impedance matching.

Reviewer 2 Report

Comments and Suggestions for Authors

The Authors have made an effort to revise the manuscript according to my recommendations, and they also provided adequate answers to many of my questions.

Unfortunately, many inconsistencies remain in the presentation of the text, such as the presentation of numerical values and units (also in the corrected and added text). For example, "d = 11μm" (no space) while in other parts of the text we read "bandwidth of 11.98 THz" (with space). The Equations are also difficult to read - see for instance Equation 8. It is, in my opinion, the responsibility of the Journal to ensure that these mistakes will not appear in the published version.

The writing is generally problematic, for example, the Abstract starts with the phrase "waves have great potential for applications in security imaging". What kind of waves?

Moreover, the Authors did not respond to how they obtained the data of Figure 2, which is another concern (albeit this Figure has been removed). 

Nevertheless, I find the technical part of the work interesting, and the proposed structure feasible, therefore I believe that the Authors should be given another chance to improve their work with another round of revisions.

Author Response

The Authors have made an effort to revise the manuscript according to my recommendations, and they also provided adequate answers to many of my questions.

Unfortunately, many inconsistencies remain in the presentation of the text, such as the presentation of numerical values and units (also in the corrected and added text). For example, "d = 11μm" (no space) while in other parts of the text we read "bandwidth of 11.98 THz" (with space). The Equations are also difficult to read - see for instance Equation 8. It is, in my opinion, the responsibility of the Journal to ensure that these mistakes will not appear in the published version.

Response:

As required, we have standardised the formatting of formulas and associated letters throughout the text; we have also ensured that figures with units have a space between them.

Update to the manuscript:

Only when d = 10 μm (pale yellow line) and d = 11 μm (green line), the overall absorptivity of the metamaterial exceeds 90% in the operating frequency range. The absorption bandwidth is 11.94 THz when d = 11 μm and 11.98 THz when d = 10 μm.

The writing is generally problematic, for example, the Abstract starts with the phrase "waves have great potential for applications in security imaging". What kind of waves?

Response:

The beginning of the abstract wants to express that terahertz waves have potential applications in security imaging.

Update to the manuscript:

AbstractTerahertz waves have great potential for applications in security imaging, wireless communication, and other fields, but efficient and tunable terahertz absorbing devices are the key to their technological development.

Moreover, the Authors did not respond to how they obtained the data of Figure 2, which is another concern (albeit this Figure has been removed).

Response:

Due to the lack of sufficient reliable data for the vanadium dioxide phase transition diagram, the original Figure 2 was merely a preliminary and inaccurate simple fitting attempt to show the trend of VO2 conductivity with temperature. Recognizing the inadequacy and potential for misinterpretation, we removed the figure.

Round 3

Reviewer 1 Report

Comments and Suggestions for Authors

I appreciate the revision of the paper. Owing to the clear explanation in the 'comments to reviewers', the advantage of the proposed design has become clear. 

Folloing is my further request for revision of the paper:

Since the broad band absorption and tramsmission is the new achievement, key design principles in your response to reviwers should be included within the paper.

Importance of combining graphen and VO2 layers should be addressed as well, which is also related to the parameters h1 and h2. 

Author Response

Reviewer 1:

I appreciate the revision of the paper. Owing to the clear explanation in the 'comments to reviewers', the advantage of the proposed design has become clear. 

Following is my further request for revision of the paper,

Since the broad band absorption and tramsmission is the new achievement, key design principles in your response to reviewers should be included within the paper.

Response:

The switch between broadband absorption and high transmissivity transmission is realised not only to fill the gap in this part of the relevant research, but also because the metamaterial device has a wide range of application scenarios in fields such as optical switching. For example, when the metamaterial is in the broadband absorption state, only the VO2 conductivity and the Fermi energy level of graphene need to be regulated to achieve the switch to the transmission mode. And VO2 phase transition response time up to nanoseconds, graphene electrical regulation for picoseconds, can be used for terahertz optical switching arrays, support for optical computing, optical interconnections and other areas of high-speed signal processing.

In addition to this, we have also included key design principles in the article.

Update to the manuscript:

Introduction

Yang et al.[24] developed a VO2-based metamaterial that achieves broadband absorption (exceeding 90%) spanning 2.81-7.71 THz. The absorptivity can be dynamically tuned from exceeding 90% to not exceeding 5% across this frequency band through VO₂ conductivity modulation. However, the existing research has achieved rich performance in adjusting the absorption rate, there are few reports on the switching between broadband absorption and transmission functions, despite its significant application potential.

Here we propose a terahertz metamaterial structure that can switch between broadband absorption and transmission using graphene and VO2 materials, due to the limited applications of metamaterial devices that currently achieve broadband absorptivity tuning. Different from the metal dielectric layer used in traditional metamaterial structures, we utilize the structure of VO2-SiO2-VO2-SiO2-Graphene (continuous layer-dielectrics-resonator layer- dielectrics-resonator layer), which is more conducive to the metamaterial realizing the function of transmission.

Design And Method

The four corners of the top graphene patterned layer consist of four quarter circles with radius r4; the pattern in the centre consists of a cross adding four half circles with radius r5 in the four directions, where the lengths of the crosses in the transverse and longitudinal directions are d. The cell period is p. The pattern in the centre consists of a cross adding four half-circles with radius r5 in the four directions.

The concentric ring VO2 structure enables multi-resonance coupling and symmetry optimization, achieving ultra-wideband, polarization-insensitive terahertz absorption. Its outer and middle ring openings form a parallel slit capacitor; when terahertz waves hit vertically with the electric field parallel to the slit, the structure acts as an LC circuit, triggering strong resonance to enhance absorption via slit electric field, metal ring eddy current loss, and Joule heat. The graphene cross-quarter-circle combination realizes "peak complementarity", enhances absorption through local electric field superposition, enables 2-4 THz broadband absorption, packs more resonators in limited area, and interacts with terahertz waves in multiple ways to expand bandwidth.

Results And Discussion

The proposed metamaterial structure can realize the mode switching between two ultra-wideband absorption and the function switching between ultra-wideband absorption and broadband transmission, with an average transmissivity of 88% in the operating frequency range, which effectively fills this research gap. This rapid switching (VO2 phase transition response time up to nanoseconds, graphene electrical regulation for picoseconds) can be used for terahertz optical switching arrays, support for optical computing, optical interconnections and other areas of high-speed signal processing.

Importance of combining graphene and VO2 layers should be addressed as well, which is also related to the parameters h1 and h2.

Response:

The purpose of combining graphene and VO2 is to break through the limitations of a single material. Graphene has the advantages of continuous fine regulation in terahertz band, very fast response and low static loss, but need to continuously apply the bias voltage to maintain the regulatory state; VO2 can be achieved through temperature regulation of the phase transition, and after the phase transition without external stimulation to maintain the state, low insulating loss, but the metal phase loss is higher, the lack of continuity of regulation. After the combination of the two, not only can VO2 achieve "coarse tuning switch", that is, from the broadband absorption of mode 1 to switch to the high transmissivity transmission of mode 3; but also with the help of graphene to complete the "fine tuning compensation", that is, the absorption bandwidth between mode 1 and mode 2 length of the adjustment. In addition, the combination of VO2 and graphene enables complementary absorption bands and ultra-broadband absorption in mode 1.

Update to the manuscript:

Graphene has the advantages of continuous fine regulation in terahertz band, very fast response and low static loss, but need to continuously apply the bias voltage to maintain the regulatory state; VO2 can be achieved through temperature regulation of the phase transition, and after the phase transition without external stimulation to maintain the state, low insulating loss, but the metal phase loss is higher, the lack of continuity of regulation. After the combination of the two, not only can VO2 achieve "coarse tuning switch", that is, from the broadband absorption of Mode 1 to switch to the high transmissivity transmission of Mode 3; but also with the help of graphene to complete the "fine tuning compensation", that is, the absorption bandwidth between mode 1 and Mode 2 length of the adjustment.

We define the frequency band range with absorption rate greater than 90% as the absorption bandwidth, and select the absorption bandwidth of VO2 in the metallic state as the main optimization goal, and the average transmittance of VO2 in the dielectric state as the secondary optimization target. After several rounds of simulation optimisation, we obtain the optimal structural parameters: r1 = 1.5 μm, r2 = 4.5 μm, r3 = 7.5 μm, r4 = 5 μm, r5 = 3 μm, a = 2 μm, b = 1 μm, c = 1.5 μm, d = 10 μm, t1 = 0.3 μm, t2 = 0.25 μm, h1 = 4.25 μm, h2 = 7.7 μm, p = 19 μm.

Reviewer 2 Report

Comments and Suggestions for Authors

The Authors made improvements and considered all my comments.
Considering that the designed, functional structure is interesting, I suggest to accept the manuscript for publication.

Author Response

Reviewer 2:

The Authors made improvements and considered all my comments.
Considering that the designed, functional structure is interesting, I suggest to accept the manuscript for publication.

Response:

Thank you very much for your positive evaluation and valuable comments. We are extremely grateful that you recognize the interest of the designed structure and recommend the publication of our manuscript. We will continue to refine our work to ensure its quality.